# Layer-specific integration of locomotion and sensory information in mouse barrel cortex

Aslı Ayaz [1,2], Andreas Stäuble[1,2], Morio Hamada[1,3], Marie-Angela Wulf[2,4], Aman B. Saleem [5] & Fritjof Helmchen [1,2]

During navigation, rodents continually sample the environment with their whiskers. How locomotion modulates neuronal activity in somatosensory cortex, and how it is integrated with whisker-touch remains unclear. Here, we compared neuronal activity in layer 2/3 (L2/3) and L5 of barrel cortex using calcium imaging in mice running in a tactile virtual reality. Both layers increase their activity during running and concomitant whisking, in the absence of touch. Fewer neurons are modulated by whisking alone. Whereas L5 neurons respond transiently to wall-touch during running, L2/3 neurons show sustained activity. Consistently, neurons encoding running-with-touch are more abundant in L2/3 and they encode the run-speed better during touch. Few neurons across layers were also sensitive to abrupt perturbations of tactile flow during running. In summary, locomotion significantly enhances barrel cortex activity across layers with L5 neurons mainly reporting changes in touch conditions and L2/3 neurons continually integrating tactile stimuli with running.

[1] Brain Research Institute, University of Zurich, Zurich, Switzerland. [2] Neuroscience Center Zurich, University of Zurich and ETH Zurich, Zurich, Switzerland. [3] Institute of Neuroinformatics, University of Zurich and ETH Zurich, Zurich, Switzerland. [4] Institute of Neuropathology, University Hospital of Zurich, Zurich, Switzerland. [5] UCL Institute of Behavioural Neuroscience, Department of Experimental Psychology, University College London, London, UK. Correspondence and requests for materials should be addressed to A.A. (email: ayaz@hifo.uzh.ch)

We sense the outside world through continuous interactions between sensory inputs and motor actions. Determining how sensory and motor information are integrated in neuronal circuits in the brain thus is critical for understanding sensory processing during behavior. Accumulating evidence suggests that in the neocortex, integration occurs at early stages of sensory processing. Several recent studies have explored the effects of locomotion on sensory processing in the visual system. These studies have found increased spontaneous and evoked activity of excitatory neurons in both primary visual cortex (V1)[1,2] and the lateral geniculate nucleus of the thalamus during locomotion[3]. These effects could not be explained by simple gain modulation or additive factors as the modulation often was not monotonic[2–4], and were also distinct for different GABAergic interneuron subtypes[5–7]. However, it remains unclear how far locomotion-related modulation of neocortical activity generalizes to other sensory modalities. Indeed, locomotion has been reported to suppress excitatory neuron activity in auditory cortex[8], suggesting that the influence of locomotion on sensory processing can be modality-specific. Because rodents heavily utilize their whiskers during navigation, characterizing the influence of locomotion on somatosensory (tactile) processing in the vibrissal system may be of particular ethological relevance.

Studies of sensorimotor integration in the vibrissa-related primary somatosensory cortex (S1 or barrel cortex) have primarily focused on whisking behavior rather than running[9–20]. For example, membrane potential fluctuations of nearby pyramidal neurons were reported to desynchronize during active whisking, without significant changes in firing rate[9,10,12]. In addition, whisking was found to modulate different interneuron classes distinctly[10,21] and to increase thalamic activity[22–24]. In contrast, the effects of locomotion on somatosensory processing have been investigated only incidentally[5,25], despite the behavioral relevance of locomotion state on somatosensory processing.

In their natural environments, rodents utilize their whiskers during navigation and the occurrence of running and active whisking are typically highly correlated[26–29]. Somatosensory processing is thus likely to be state-dependent: during locomotion, mice need to monitor their location while remaining sensitive to sudden changes in the environment, such as encountering a large obstacle. During stationary periods on the other hand, mice may emphasize subtler aspects of tactile sensations such as the texture or shape of surfaces.

It is likely that locomotion exerts distinct modulations on neurons in different cortical layers. In barrel cortex, the input and output connections are highly layer-specific. For example, afferent inputs to barrel cortex from the ventral posterior medial (VPM) nucleus in thalamus mainly target L4 (plus L5/6; c.f. see ref. [30]), whereas axons from the posterior medial (POM) nucleus terminate in L1 and L5A[31–33]. Long-range projections between S1 and other cortical areas, e.g., vibrissal motor cortex (M1) and secondary somatosensory cortex (S2), also exhibit substantial laminar specificity[32,34–36]. Laminar-specific connectivity suggests different functional roles of superficial and deep-layer neurons in S1. In line with this notion, recent studies that directly compared sensory processing across layers have started to uncover functional differences between cortical laminae[18,25,37–40]. In visual cortex, most studies investigated locomotion effects in superficial (supragranular) layers[1,5,41,42] and only few examined deeper (infragranular) layers[2,4]. A comparison of locomotion-related modulation of cortical activity across layers is missing thus far.

Here, we investigate whether L2/3 and L5 neurons of barrel cortex are modulated during running, and whether they differ in their integration of sensory and motor signals. We use two-photon calcium imaging in head-restrained mice running along a virtual tactile wall. This setting allows us to study how two motor behaviors—whisking and running—are integrated with touch events and prolonged ongoing sensory touches. We find that running strongly modulates ongoing neuronal activity and touch-evoked responses in barrel cortex, and that this modulation cannot be explained by the accompanying whisking behavior. Furthermore, we reveal distinct features of L2/3 and L5 neurons regarding the representation of continuous touch stimuli and the integration of locomotion and touch.

## Results

**Calcium imaging of L2/3 and L5 in a tactile virtual reality**. To study the effects of locomotion on touch processing in barrel cortex, we built a tactile virtual reality setup to mimic running and whisking along a wall in the dark (Fig. 1a; see Methods; Supplementary Video 1). Mice were head-fixed on top of a ladder wheel under a two-photon microscope. We continually recorded run speed and induced touches by bringing a sandpaper-covered rotating cylinder ('textured-wall') in contact with the whiskers on one side of the face. Whisker movements were imaged with a high-speed camera (Supplementary Video 2). This setup enabled us to consider several experimental conditions (Fig. 1b): First, the mouse was free to run or rest on the treadmill in the absence of the textured-wall ('No-touch'). Second, the texture was moved in contact with the whiskers after the mouse had run a predefined distance, rotating at the same speed as the animal's run speed ('Closed-loop'). Third, texture speed and run speed were decoupled ('Open-loop'). The varying combinations of texture and run speed in this 'Open-loop' condition allowed us to dissect their respective contributions to neuronal activity in S1. Finally, to explore neuronal responses to an abrupt mismatch of tactile flow and run speed, we applied brief perturbations during Closed-loop trials by halting the rotating texture for 2 s at a random time point during the touch. In all conditions, mice were free to move on the treadmill and did not receive any rewards. To avoid whisker stimulation by the treadmill, the bottom rows of whiskers were trimmed. Measurements were made in several experimental sessions spread over 3 weeks (see Methods).

To measure neuronal activity across cortical layers, we injected AAV2.1-EFα1-R-CaMP1.07 into barrel cortex of adult mice, resulting in expression of the genetically encoded calcium indicator R-CaMP1.07[43,44] in L2/3 and L5 neurons (Fig. 1c; see Methods). In line with a previous report[45], only few GABAergic neurons expressed R-CaMP1.07 with this virus construct (Supplementary Fig. 1). Thus, R-CaMP1.07-expressing cells mainly represent pyramidal neurons. Based on the lack of expression in L4 (Fig. 1c and Supplementary Fig. 2) we selected imaging areas either above or below this laminar landmark (six areas each in L2/3 and L5; imaging depths 221–385 μm for L2/3 and 450–664 μm for L5; $n = 5$ mice). In total, we measured calcium signals in 426 neurons in L2/3 and 275 neurons in L5 repeatedly over several days. Figure 1d, e shows example calcium transients measured in L2/3 and L5 subpopulations, respectively. In acute brain slice experiments we verified that pyramidal neurons in L2/3 and L5 display similar action potential-evoked calcium dynamics (Supplementary Fig. 3; see Methods). Measurements from L2/3 and L5 neurons were performed separately in distinct imaging sessions.

**Running modulates barrel cortex activity more than whisking**. We first investigated how locomotion and whisking affect the activity of S1 neurons in the absence of texture touch (No-touch sessions). Based on the recorded running speed and measured whisker movements, we defined four different behavioral states (Fig. 2a; see Methods): Animals spent about half of their time running, which was always accompanied by simultaneous

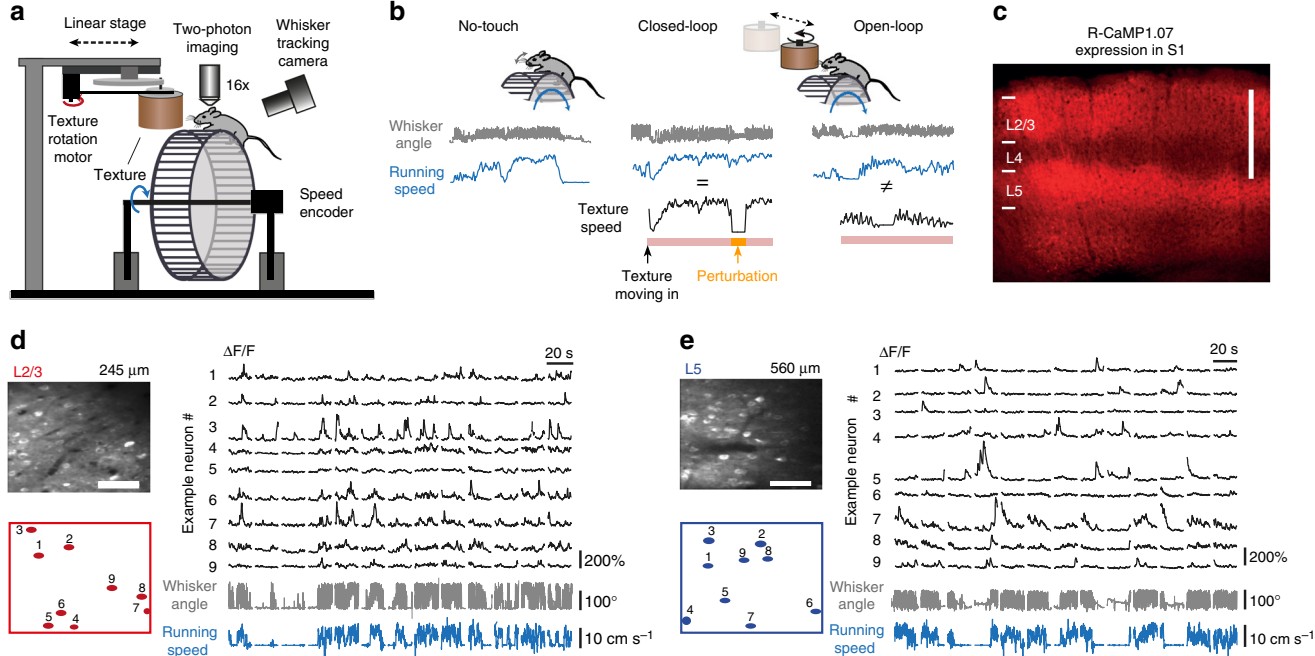

**Fig. 1** Calcium imaging in L2/3 and L5 of mouse barrel cortex during various running and whisking conditions. **a** Schematic of virtual reality setup with a head-restrained mouse on top of a rung-ladder treadmill. A sandpaper-covered cylinder (texture) can be moved in contact with the whiskers. Run speed is tracked with an encoder and the mean whisker angle is monitored with a high-speed video camera. **b** Example traces of whisker angle (gray), run speed (blue), and texture-rotation speed (black), illustrating the three experimental conditions: 'No-touch', 'Closed-loop', and 'Open-loop'. The pink bottom bar indicates when the texture contacts the whiskers. The orange segment highlights the intermittent halt of texture-rotation to introduce a brief perturbation period in Closed-loop trials by uncoupling run speed and texture speed. **c** Confocal image of virally-induced R-CaMP1.07 expression pattern in a coronal slice of somatosensory cortex of a wild-type mouse. Scale bar is 500 μm. **d** Left: In vivo two-photon image of R-CaMP1.07-expressing L2/3 neurons with selected ROIs below (scale bar, 50 μm). Right: ΔF/F calcium transients of nine example neurons with simultaneously recorded mean whisker angle (gray) and running speed (blue) below. **e** Same as in (**d**) but for example L5 neurons in S1

whisking as reported previously[26]. Stationary periods included whisking as well as no-whisking episodes whereas animals almost never ran without whisking. To assess activity changes induced solely by whisking, we excluded all running periods and compared the mean fluorescence change during episodes of whisking and no-whisking for each neuron. Whisking barely modulated the mean activity in L2/3 and L5 populations (Fig. 2b and Supplementary Fig. 4). To account for differences in activity levels we calculated a whisking modulation index (MI) for each neuron (as defined in ref. [6], see Methods). Whisking MIs were close to but significantly different from zero ($p = 5.3 \times 10^{-11}$ for L2/3 and $p = 3.1 \times 10^{-7}$ for L5, Wilcoxon signed rank test) and not different for L2/3 and L5 neurons (Fig. 2c; $0.071 \pm 0.009$, $n = 342$, and $0.055 \pm 0.013$, $n = 168$, respectively; mean ± s.e.m.; $p = 0.117$, Wilcoxon rank sum test). As an alternative analysis, we averaged neuronal fluorescence changes aligned to onsets of whisking while the animal was stationary (considering only fluorescence trace segments until whisking stopped). Consistent with a weak modulation by whisking, only a minor fraction of neurons (3% in L2/3 and 8% in L5) showed significant fluorescence increases upon whisking onset (Fig. 2d and Supplementary Fig. 4).

Similarly, we examined how running modulates activity of S1 neurons. As animals almost always whisked during locomotion (Fig. 2a; cf. ref. [26]), any running period also included whisking (referred to as 'running/whisking'). For each neuron, we compared the mean activity during running/whisking and during resting periods (including both 'whisking' and 'no-whisking' episodes). Running/whisking caused significant mean ΔF/F increases in the majority of both L2/3 and L5 neurons and decreases in some neurons (Fig. 2e and Supplementary Fig. 4). Consistently, a running MI—defined in analogy to the whisking

MI—revealed increased activity during running/whisking for both populations (Fig. 2f; running MI $0.210 \pm 0.016$, $n = 342$, for L2/3 and $0.329 \pm 0.023$, $n = 168$, for L5 neurons, respectively; $p < 10^{-20}$ for both populations, Wilcoxon signed rank test) with significantly larger running modulation for L5 neurons ($p = 1.2 \times 10^{-5}$, Wilcoxon rank sum test). We also computed average fluorescence changes aligned to locomotion onsets, considering only fluorescence trace segments until running stopped. About one-third of neurons showed a significant increase in activity upon running onset (32% and 38% for L2/3 and L5, respectively; Supplementary Fig. 4). This increase persisted over several seconds as reflected in the population average of run-onset-aligned responses of L2/3 and L5 neurons (Fig. 2g). Taken together, the No-touch sessions revealed that running/whisking increases the activity of both L2/3 and L5 neurons in barrel cortex to a larger extent than whisking alone.

We also examined the effect of varying run speed on the activity of barrel cortex neurons in the absence of sensory stimulation. In total, 39% of L2/3 and 45% of L5 neurons were modulated by run speed ($p < 0.001$; ANOVA for run speed $>1$ cm s$^{-1}$; Supplementary Fig. 5; see Methods). Half of these neurons showed band-pass tuning (50% and 54% for L2/3 and L5, respectively), such that their mean calcium signal was highest at a specific intermediate run speed but reduced at lower and higher speeds. Smaller fractions of run-speed modulated cells monotonically increased (19 and 10%) or decreased (31 and 36%) their responses with increasing run speed.

**Differential responses to wall touch in L2/3 and L5 neurons.** After determining the effect of running and whisking in the absence of sensory stimulation, we next compared responses

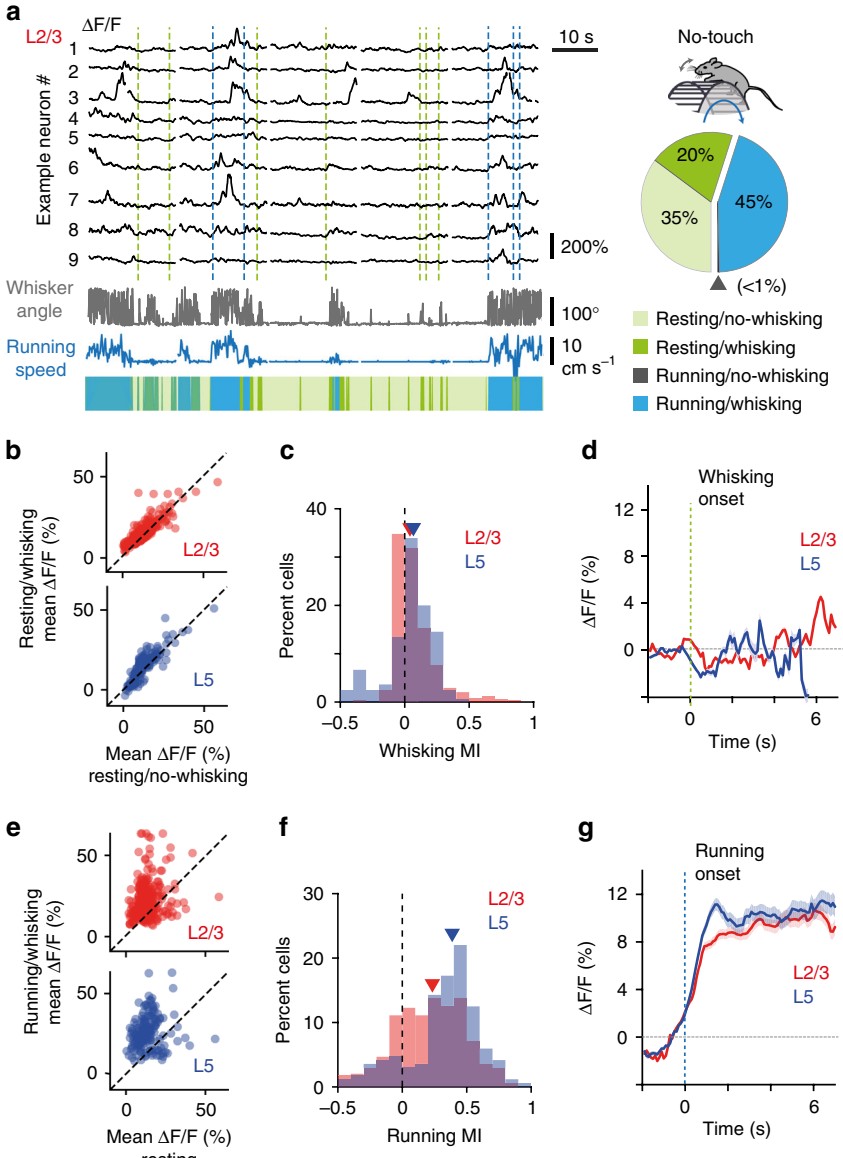

**Fig. 2** Running with concomitant whisking increases L2/3 and L5 activity more than whisking alone. **a** Left: Example $\Delta F/F$ traces along with whisker angle and running speed in the initial No-touch trials of Fig. 1d. The four possible running and whisking state conditions are color-coded. Green and blue vertical dotted lines mark whisking onset (only during resting periods) and running onset, respectively. Right: Pie chart of the distribution of times spent in the four states for five mice (in the 'No-touch' condition). Gray triangle indicates <1% fraction of time spent in the running/no-whisking state. **b** Scatter plots of mean $\Delta F/F$ amplitude in resting/whisking periods versus resting/no-whisking periods for 342 L2/3 neurons (red) and 168 L5 neurons (blue). Dashed lines indicate unity lines. **c** Distribution of whisking modulation index (MI) for L2/3 and L5 neurons. Red and blue triangles indicate medians. **d** Population average of whisking-onset-aligned $\Delta F/F$ traces for all recorded neurons. **e** Scatter plots of mean $\Delta F/F$ amplitude in running/whisking periods versus resting periods for L2/3 (top) and L5 (bottom) neurons. **f**, **g** Analog plots to (**c**), (**d**) for running modulation index (MI) and running-onset-aligned $\Delta F/F$ traces

evoked by texture touch in the Closed-loop condition. In each trial, after the mouse had run a predefined distance, a textured wall moved in contact with the whiskers, rotating at the same speed as the animal was running (Fig. 3a). We aligned the fluorescence traces to 'touch onset', defined as the time point when the texture started moving toward the whiskers (see Methods). More than half of the neurons were responsive to wall touch (64%, 268/420, in L2/3 and 57%, 157/275, in L5) and these responses were not induced by the sound of the texture-rotation motor or the linear stage (see Methods). Both L2/3 and L5 neurons displayed a strong initial response to touch. However, L2/3 neurons continued to exhibit a sustained response to ongoing wall touch whereas L5 neurons showed a transient response (Fig. 3b–e). Although a rough-textured wall elicited

slightly larger activity than a smooth-textured wall in both L2/3 and L5 neurons, the difference in temporal response profile remained the same, independent of texture identity (Supplementary Fig. 6). Because action-potential-evoked somatic calcium dynamics is fast and similar for L2/3 and L5 neurons (Supplementary Fig. 3; ref. [46]), the difference in transient versus sustained touch responses of L5 and L2/3 neurons does not reflect differences in calcium dynamics, but rather points to differences in their local and long-range connectivity governing their synaptic inputs and action potential firing patterns. Consistent with this notion, no difference in temporal profile was observed for locomotion onset responses, with calcium signals showing sustained activation for both L2/3 and L5 neurons (Fig. 2g and Supplementary Fig. 7).

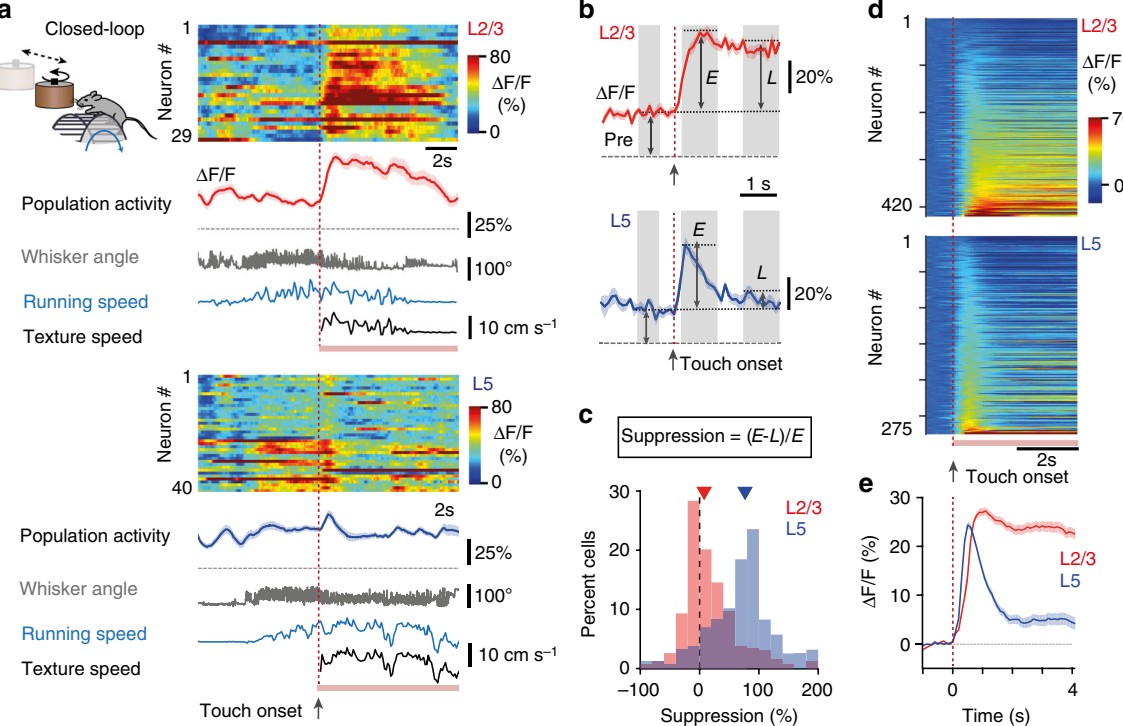

**Fig. 3** Wall touch during running evokes sustained responses in L2/3 and transient responses in L5. **a** Population dynamics in L2/3 (left) and L5 (right) neurons for two example trials under the Closed-loop condition. The heat map represents $\Delta F/F$ calcium transients in pseudo-color code. Below the heat maps the population mean $\Delta F/F$ trace (solid line, ±s.e.m.), whisker angle (gray), running speed (blue), and texture speed (black) are shown. Touch onsets are indicated by red vertical dotted lines; pink bars represent periods of texture contact. **b** Mean $\Delta F/F$ traces aligned to touch onset (red dotted line) for an example L2/3 and L5 neuron. Shaded areas indicate windows for analysis (pre-touch, 'Pre', −1 to −0.3 s; early, 'E', 0.3 to 1.3 s; late, 'L', 2 to 3 s). **c** Distribution of percent suppression as defined in the equation for L2/3 and L5 touch-responsive neurons. Arrowheads mark medians of 76.6% and 7.7%, respectively. **d** Heat maps representing touch onset responses of all L2/3 (top, $n = 420$) and L5 (bottom, $n = 275$) neurons in Closed-loop condition. Pre-touch activity was subtracted from calcium transients and neurons were sorted according to their $\Delta F/F$ responses in the early window, E. **e** Population averages (±s.e.m.) of touch-aligned $\Delta F/F$ traces for all touch-responsive neurons in L2/3 (red, $n = 268/420$) and L5 (blue, $n = 157/275$). Neurons were considered touch-responsive when $E > 2\sigma$, where $\sigma$ is the minimum standard deviation (see Methods)

We further evaluated the distinct temporal profiles of L2/3 and L5 responses. In an early response window, the peak $\Delta F/F$ change relative to the mean $\Delta F/F$ value in a pre-touch period was $32 \pm 0.9\%$ and $26 \pm 0.9\%$ for responsive L2/3 and L5 neurons, respectively (Fig. 3b; mean ± s.e.m.; $p < 10^{-4}$, Wilcoxon rank sum test). In a late response window (2–3 s after touch onset) the $\Delta F/F$ traces showed a strong decrease in L5 but not in L2/3, which we quantified by calculating the $\Delta F/F$ suppression in the late versus the early window. Suppression was significantly larger for L5 compared with L2/3 neurons (Fig. 3c; $69.6 \pm 4.4\%$ for L5 and $16.6 \pm 2.7\%$ for L2/3, $p < 10^{-4}$, Wilcoxon rank sum test), reflecting again the difference in transient versus sustained responses. In addition, touch modulation of $\Delta F/F$ signals was inversely correlated to locomotion modulation, especially for L5 neurons (Supplementary Fig. 8).

In the Closed-loop condition, the wall touch almost always happened during running as mice had to travel a certain distance before reaching a wall. To test whether laminar specificity of wall touch responses depends on the locomotion state, we also examined touch responses in Open-loop condition (Fig. 4a). This allowed us to sort wall touch events into two groups according to whether the touch occurred during running or during resting (see Methods). Although neuronal activity overall was lower during resting, L2/3 neurons gave a sustained response after wall touch in either condition, whereas L5 neuron activity decreased over time (Fig. 4b–g and Supplementary Fig. 8). In summary, our findings reveal that touch-evoked activity in barrel cortex shows laminar specificity, with L2/3 neuronal subsets showing sustained activity

during continuous touch whereas subsets of L5 neurons respond transiently.

**Responses to mismatch of running speed and tactile flow.** In visual cortex, a subset of neurons report mismatches between the animal's motion and visual flow by increased activity[41,42]. Here, we explored whether mismatch-sensitive neurons also exist in barrel cortex. In the Closed-loop condition, we introduced strong mismatches between locomotion and tactile flow by abruptly halting the rotating texture for 2 s at a random time point during the touch. We found small numbers of neurons that either increased (up-modulated) or decreased (down-modulated) their activity upon such perturbation (example cells shown in Fig. 5a, b). For quantification, we averaged $\Delta F/F$ traces after alignment to the perturbation onset, and defined the perturbation response amplitude as the mean $\Delta F/F$ change in a 2-s time window after perturbation relative to pre-perturbation baseline (Fig. 5b). The distribution of perturbation response amplitudes was similar for L2/3 and L5 neurons, albeit with a slightly but significantly lower mean value for L2/3 neurons (Fig. 5c; $-4.2 \pm 0.6\%$, $n = 420$ and $-1.7 \pm 0.7\%$, $n = 275$, for L2/3 and L5, respectively; $p < 0.001$, Wilcoxon rank sum test). Overall, 4.5% (19/420) of L2/3 and 3.3% (9/275) of L5 neurons showed a significant increase in response to mismatch between running speed and tactile flow (see Methods). Significantly down-modulated neurons were more abundant in L2/3 (12.4%, 52/420) than in L5 (6.6%, 18/275) (Fig. 5d). Decreased activity was the dominating effect at the population

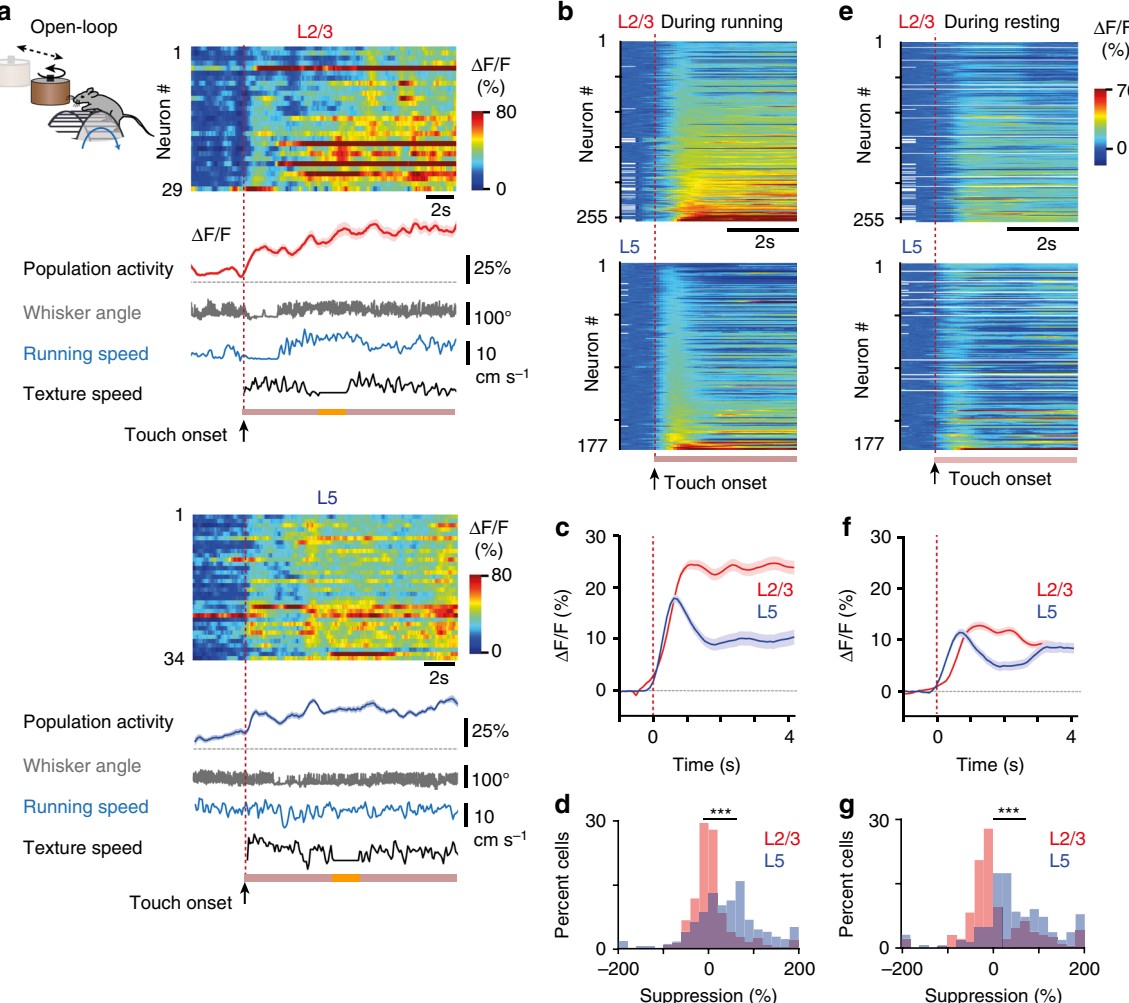

**Fig. 4** Touch-evoked responses under Open-loop condition. **a** Population dynamics in L2/3 (left) and L5 (right) neurons for two example trials under Open-loop condition. The heat map represents ΔF/F calcium transients in pseudo-color code. Below the population mean ΔF/F trace (solid line, ±s.e.m.) are whisker angle (gray), running speed (blue), and texture speed (black). Touch onsets are indicated by red vertical dotted lines; pink bars represent periods of texture contact; orange segment is the period where texture rotation is briefly stopped. Imaging field of views are the same as in Fig. 3a, with identical neuron numbers for L2/3. In L5 some neurons were not captured in the corresponding imaging session, hence the smaller number of neurons. **b** Of the touch-responsive neurons in Open-loop condition 255 of 342 L2/3 neurons and 177 of 236 L5 neurons had touch events during running. Heat maps show touch onset responses of these neurons in L2/3 (top) and L5 (bottom) during running. Initial white segments correspond to NaN values for neurons with very short pre-touch periods. **c** Population averages (±s.e.m.) of touch-aligned ΔF/F traces during running. **d** Distribution of percent suppression of touch responses during running. Mean suppression was 8.1 ± 2.9% and 44.1 ± 5.6% for L2/3 and L5 neurons, respectively ($p = 1.3 \times 10^{-16}$, Wilcoxon rank sum test). **e** Heat maps show touch responses that occurred during resting for the same neurons shown in (**b**). Neurons that did not have touch events during resting conditions are shown as white rows in heat maps. **f** Population averages (±s.e.m.) of touch-aligned ΔF/F traces during resting (245/342 L2/3 neurons, red, and 167/236 L5 neurons, blue). **g** Mean suppression was 6.4 ± 4.4% for L2/3 and 49.8 ± 6.3% for L5 neurons, respectively ($p = 4.0 \times 10^{-14}$, Wilcoxon rank sum test)

level both in L2/3 and L5 (Supplementary Fig. 9). While such decrease in activity may be explained by reduced stimulation of the whiskers when the rotation of the texture cylinder is stopped, the increased activity upon texture halt can be considered as representing a true mismatch signal. Our Open-loop experiments also revealed that increased mismatch responses were not a pure sensory response: neurons did not show increased activity to texture stall when animals were at rest (Fig. 5e, f), in addition the number of mismatch responsive neurons during running was similar to Closed-loop condition (Fig. 5g).

**More-prominent integration of locomotion and touch in L2/3.** After revealing neuronal population responses to salient events (e.g., running onset, touch, and perturbation), we evaluated how homogenously these sensorimotor aspects are functionally

represented among L2/3 and L5 neurons in barrel cortex. To this end, we conducted Open-loop experiments during which various combinations of running and touch occurred. Some neurons faithfully increased activity when the animal touched the wall, independent of running ('Touch cells'); another subset of neurons was most active when the animal was running independent of the wall touch ('Run cells'); a relatively large population of neurons was most responsive when wall touch and running occurred concomitantly ('Integrative cells'); finally, some rare neurons were most active when the animal was stationary in the absence of wall touch ('Rest cells') (Fig. 6a). To classify neurons according to these categories, we computed the mean ΔF/F values for the four combinations of locomotion and touch state ('no-running/no-touch', 'running/no-touch', 'no-running/touch' and 'running/touch') and categorized all neurons

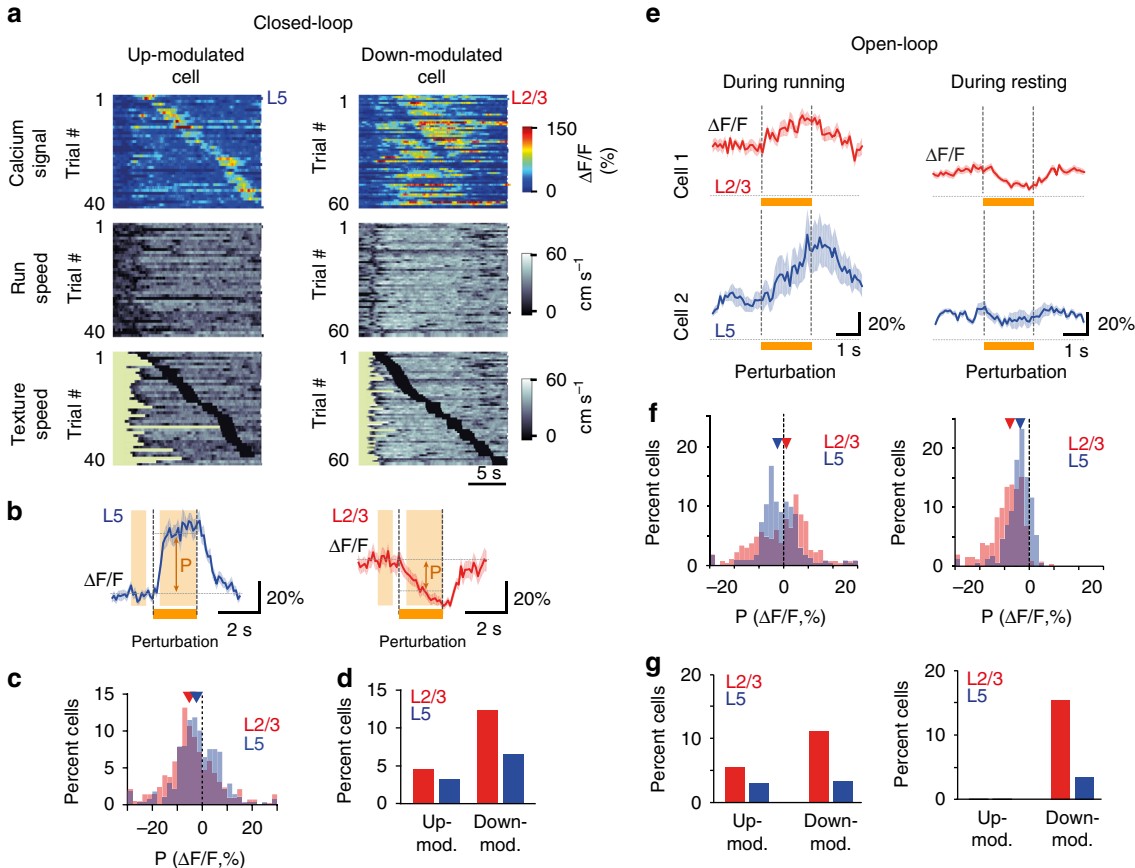

**Fig. 5** A subset of neurons responds to perturbations of tactile flow. **a** Calcium signals for two example neurons across multiple Closed-loop trials. Trials are sorted according to when the perturbation (2-s texture halt) occurred. Running speed and texture speed are plotted below with black periods marking stop of texture rotation. The left L5 neuron is an example of an up-modulated cell whereas the right L2/3 neuron exemplifies down-modulation upon perturbation. **b** Mean $\Delta F/F$ traces (±s.e.m.) aligned to perturbation onset for the example neurons in (**a**). '$P$' indicates the amplitude of perturbation-induced modulation defined as the difference between mean $\Delta F/F$ values before (−1 to −0.3 s window) and during perturbation (+0.3 to 2 s, orange shaded areas). **c** Distribution of perturbation-induced modulation for all L2/3 (red, $n = 420$) and L5 (blue, $n = 275$) neurons. Triangles mark medians of −5.1% and −2.3% for L2/3 and L5 neurons, respectively. **d** Percentage of significantly up-modulated ($P > 2\sigma$) and down-modulated ($P < −2\sigma$) neurons in L2/3 and L5 populations, respectively, with $\sigma$ denoting baseline noise (see Methods). **e** Perturbation responses of two example neurons in Open-loop condition, where sudden stalling of the texture rotation may occur during running (left panels) or resting (right panels). **f** Distribution of perturbation modulation ($P$) for neurons in L2/3 (red, $n = 269$) and L5 (blue, $n = 234$) neurons during running (left panel, mean $P$: −0.7 ± 0.7% and −1.7 ± 0.6% for L2/3 and L5; $p < 0.05$) and during resting (right panel, 253 L2/3 neurons, red, and 172 L5 neurons, blue; mean $P$: −9.2 ± 0.5% and −4.3 ± 0.4% for L2/3 and L5, respectively, $p < 0.0001$). Wilcoxon rank sum test was used for distribution comparisons. **g** Percentage of neurons that are significantly up-modulated ($P > 2\sigma$) and down-modulated ($P < −2\sigma$) for L2/3 (red) and L5 (blue) populations during running (left) and during resting state (right)

according to their highest response (see Methods). 77.2 ± 9.6% of L2/3 neurons were integrative cells compared with only about half of the neurons in L5 (51.2 ± 4.5%). Accordingly, the two subpopulations of touch cells and run cells were larger in L5 than in L2/3 (Fig. 6b, c). At the population level, mean $\Delta F/F$ values of L2/3 neurons were similar for running/no-touch and no-running/ touch conditions with an approximately linear summation for concomitant running/touch (Fig. 6d). In comparison, mean $\Delta F/F$ values of L5 neurons were larger for running-only compared with touch-only condition, with only a marginal increase in population activity when wall touch occurred during running.

After categorizing cells according to their responses in the presence or absence of motor and sensory inputs, we also tested how well time-variant information about motor and sensory variables explained neuronal responses. We applied a random forest algorithm to predict calcium signals from texture-rotation speed, run speed and whisking envelope during wall touch periods in 'Open-loop' experiments, where these three variables were independent (Supplementary Fig. 10; see Methods). We found that run speed contributed the most to predicting neuronal

responses, as prediction quality showed the largest decrease when this parameter was shuffled. Texture speed explained a smaller but significant proportion of the variance whereas shuffling whisking envelope did not affect prediction quality. These findings hold for both L2/3 and L5 populations, with overall prediction quality being slightly lower in L5 than in L2/3.

The larger fraction of integrative neurons in L2/3 and their enhanced mean activity during concurrent wall touch and running suggest a more effective integration of sensory and motor information in superficial layers. To test this hypothesis we compared for each neuron mutual information between the best predictive variable, run speed, and the calcium signal during no-touch and wall-touch periods of the Open-loop condition (see Methods). At the population level mutual information was increased in the presence of wall touch for superficial neurons but decreased for L5 neurons (Supplementary Fig. 10). Although integrative cells increased their mutual information in both L2/3 and L5, the abundance of integrative neurons in superficial neurons explained this difference between layers. Taken all together we conclude that L2/3 neurons show a stronger

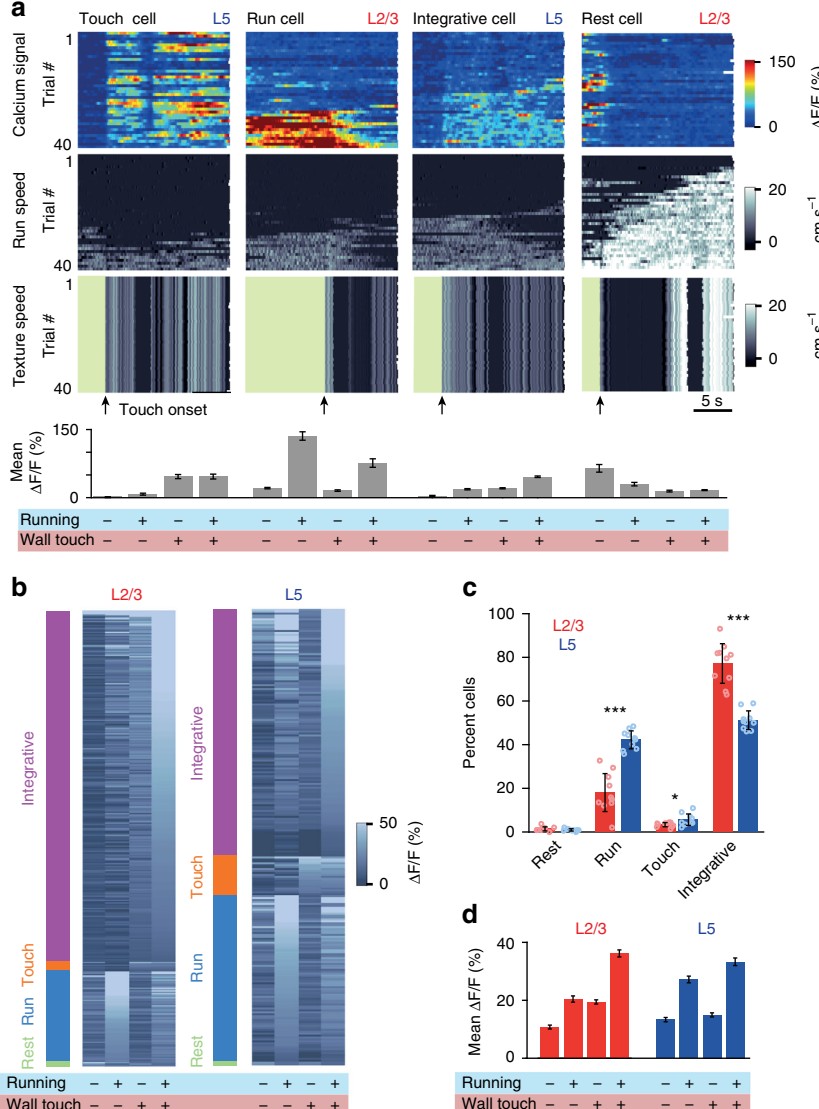

**Fig. 6** A higher fraction of neurons integrate locomotion and concurrent wall touch in L2/3 compared with L5. **a** Four example neurons with different response properties recorded during Open-loop stimulation. Each column presents data from a single neuron across trials in a single session. Top panel heat map shows ΔF/F calcium signal with each row representing a trial. Trials are sorted according to the mean run speed of a trial. Middle and bottom panels show the run speed and texture-rotation speed of corresponding trials. Green periods in the bottom panel indicate when the texture was not in contact with whiskers. Bar graphs at the bottom are mean ΔF/F activity during the four stimulus and movement conditions: no-touch/no-running, running/ no-touch, touch/no-running, and touch/running. **b** Categorization of all L2/3 ($n = 338$, 5 mice) and L5 ($n = 236$, 4 mice) neurons according to their activity during their first Open-loop session. Color bars on the left show categories, gray scale shading shows mean ΔF/F at four stimulus/behavior conditions. Cells are sorted according to their mean activity during 'running/wall touch' condition relevant to their assigned category. **c** Normalized distributions of four response categories in L2/3 and L5 populations. Means and standard deviations are calculated from 10 random selections of 11 sessions out of total 24 recording sessions. (Two-sampled T-test is performed to compare distributions for each category $p_{rest} = 0.2$, $p_{run} = 5.7 \times 10^{-7}$, $p_{touch} = 0.02$, $p_{integrative} = 3.7 \times 10^{-7}$.) **d** Average population responses for the four stimulus/movement conditions for L2/3 (in red) and L5 (in blue) neurons shown in (**b**) (mean ΔF/F ± s.e.m.)

integration of locomotion and sensory information than neurons in L5.

## Discussion

Using a tactile virtual reality setup we found that neuronal activity across barrel cortex layers is strongly increased by running, more than by whisking. Furthermore, we found that continuous wall touch evokes differential responses across layers, transient ones in L5 neurons and sustained ones in L2/3 neurons. In line with this finding, the fraction of neurons best coding for running-with-touch was larger in L2/3 compared with L5.

Moreover, in both L2/3 and L5 small subsets of neurons were sensitive to sensorimotor mismatches. Taken together, our findings highlight the strong influence of locomotion on barrel cortex activity and reveal a layer-dependence of touch responses during active locomotion.

Accumulating evidence suggests that running strongly modulates sensory processing, but effects vary across sensory modalities and among cell types (for reviews, see refs. [47,48]). We found that S1 neurons increase their activity during running (accompanied by whisking). In the absence of sensory stimulation, about 30% of both L2/3 and L5 neurons displayed a

significant run-onset responses, consistent with a previous study reporting similar effects mainly from L4 and L5 neurons[25] as the number of extracellularly recorded supragranular neurons in their study were limited. We also characterized S1 neurons to have tuned responses to run speeds. The distribution of speed tuning properties in S1 are similar to what has been described in V1[4], with majority of neurons being tuned to an intermediate run speed and smaller number of neurons to have monotonically increasing and decreasing responses to increasing run speeds. Another study on somatosensory processing used running as a means to make mice continuously whisk against a stimulus bar, but did not report on neuronal activity changes due to running[40].

In our experiments, whisking minimally increased barrel cortex activity, and we found very few whisking-onset responsive cells in both superficial and deep layers. Previous studies have reported inconsistent results regarding modulation of barrel cortex activity by whisking: some have reported changes in membrane potential dynamics but not firing rates[9,37,49], while others have reported prominent increases in firing rate upon whisking[37,38]. Our findings are more consistent with the former results, but further dissection of subtype-specific modulations in both superficial and deep layers is required. Other studies characterized whisking neurons depending on how well they encoded whisking dynamics such as whisker angle and the curvature compared with touch events[18]. They classified about 17% of superficial neurons and L5A apical dendrites to code for whisking. However this study was conducted in task-performing animals with a single intact whisker, both of which can affect neural responses. In our study, we did not have access to the dynamics of single whiskers.

Our results thus provide evidence that locomotion exerts strong effects on sensory processing in barrel cortex. One may therefore have to rethink the commonly applied notion that barrel cortex employs sparse coding[50,51]. The increased activity observed both in superficial and deep layers during running suggests that barrel cortex may employ state-dependent encoding strategies. In an ethologically relevant situation such as navigation, increased activity might provide more efficient and faster sensory coding.

Touch-evoked responses during running involved initial strong activation of both L2/3 and L5 neurons, but only in L2/3 activity remained elevated during continuous stimulation. This difference in sustained versus transient response profiles of superficial and deep-layer neurons, respectively, may reflect different functional roles of layers during active behavior. Superficial cortical layers appear to 'stay online' during ongoing sensory-motor sampling, presumably to continually monitor the interactions with the world and match external stimuli to internal expectations. In contrast, deeper layers seem to react to salient changes of the behavioral context. Via their output projections, L5 neurons may convey this information to relevant subcortical regions, such as thalamus, striatum, and brain stem, in order to promptly and accordingly adapt the animal's behavior.

One explanation for such discrepancy in temporal response profiles between L2/3 and L5 could be intrinsic differences of neuronal classes. Indeed, several studies reported differences in adaptation properties of subtypes of pyramidal cells. For example, a recent study revealed distinct membrane depolarizations in S2- and M1-projecting L2/3 neurons: while S2-projecting neurons robustly signaled sensory information during repetitive active touch, M1-projecting neurons displayed strongly adapting post-synaptic potentials[52]. In our experiments, a predominant activation of a larger population of S2-projecting neurons could explain the observed sustained response in L2/3. Similarly, L5A neurons, which mainly send projections to other cortical areas, have an adapting spiking pattern whereas L5B neurons, whose projections

target subcortical areas, have a regular spiking pattern upon current injection[53–55]. The majority of our L5 imaging fields of views were localized around L5A. Thus, adapting responses of these cells might explain the observed transient activity upon continuous texture stimulation. On the other hand, electro-physiological recordings in rat barrel cortex while stimulating whiskers at different frequencies showed no significant difference in adaptation responses of superficial and deep-layer neurons[56]. This indicates that sustained versus transient responses may result not only from intrinsic differences of L2/3 and L5 neurons, but in addition, might involve circuit mechanisms on the local as well as long-range scale.

Inhibitory neurons within the local network might mediate differential response profiles across lamina. For instance, Pluta et al.[40] reported that activation of L4 neurons, which in turn recruited a population of inhibitory fast spiking neurons in L5, caused suppression of L5 neurons but increases in activity of L2/3 neurons. In this study, mice were continuously running. Our results are in line with this observation and in addition suggest that recruitment of inhibition might be enhanced due to the running state of the mice because suppression was smaller when the animals were stationary. Another plausible mechanism to explain the late suppression of L5 neurons is the recruitment of frequency-dependent disynaptic inhibition (FDDI) of inhibitory Martinotti cells[57]. In addition to L5 pyramidal cell inputs, L5 Martinotti cells receive strong input from L2/3 neurons[58–61]. Considering the more sustained activity of L2/3 neurons upon continuous wall-touch, interlaminar recruitment of L5 Martinotti cells might underlie suppression of L5 pyramidal neurons[61]. In addition, this inhibition through Martinotti cells is initially weak but facilitated with continuous input[62], thereby possibly accounting for the delayed suppression of L5 responses. Several studies have also suggested that locomotion effects are mediated via disinhibition of pyramidal neurons by activation of vasoactive intestinal peptide (VIP) expressing interneurons upon locomotion[5,6]. VIP neurons are more numerous in superficial layers, albeit their axons can reach different layers. Hence, their activation might lead to more effective disinhibition in L2/3 upon running compared with L5, creating a disparity in responses of superficial and deep-layer neurons.

Finally, distinct long-range afferent inputs to L2/3 versus L5 neurons in general could underlie the distinct response profiles[32,34–36]. For instance, POM sends axons to L1 and L5[31,33]. Although the specificity of targeting of L1 axons onto apical dendrites of both L2/3 and L5 neurons remains unclear, specific inputs to L5A could explain distinct response patterns. To our best knowledge, changes to POM activity upon running remain to be investigated. In addition, layer-specific inputs originating from motor cortex to somatosensory cortex may contribute to response diversity.

An important aspect of our study is that it provides insight about sensory-motor represenations across cortical layers. Laminar organization of a cortical column in the mammalian neo-cortex is highly preserved throughout evolution and it is of key importance to compare response properties across layers to better understand their functional role. One suggested role of laminar organization in cortex is that it may provide a framework for hierarchical processing such that a 'higher' cortical area sends expectation information through axonal projections to superficial neurons of a 'lower' sensory area, where top-down inputs are integrated or compared with bottom-up inputs[63,64]. According to this model, one might expect expectation mismatch signals to be more prominent in superficial than in deep layers. In this study, we indeed found evidence of mismatch responses in barrel cortex but it was not limited to superficial layers. In our experiments, around 5% of both superficial and deep-layer neurons in

S1 showed increased activity to a mismatch between the animal's running speed and texture speed. Abrupt stalling of tactile flow while the animal was stationary did not cause increased activity, suggesting that the perturbation responses we observed were not driven solely by changes in sensory parameters but required integration with the animal's motor state. A similar study in primary visual cortex reported that a comparable subset (13%) of L2/3 neurons respond to expectation mismatch in visual stimuli, a brief stalling of the visual stimulus flow while the animal is running on a treadmill[42]. They also reported mismatch responses at the population level, different from our experimental results showing a decrease in population activity upon mismatch, which can be explained by a larger fraction of neurons decreasing their activity upon texture-rotation stall in S1. To our best knowledge, there is no comparison of mismatch responses of L2/3 neurons with that of L5 neurons in visual cortex. Mismatch responses in deeper layers do not disprove the current model of predictive processing, but do provide insight into how these signals might be relayed to other brain areas. Dissection of target-specific sub-populations of deep-layer neurons will be necessary to further reveal how mismatch signals are processed throughout the brain. In addition, the contribution of different subtypes of inhibitory interneurons in shaping the response dynamics of neuronal populations in superficial versus deeper cortical layers during sensorimotor integration warrants further investigation.

Within the local microcircuitry of barrel cortex, we found a distributed representation of sensory information (textured wall) and motor information (running). The majority of neurons, especially in L2/3, responded most strongly during the combined condition of run-with-touch. In L5, fewer neurons exhibited integrative properties whereas half of the neurons responded best to a single modality (wall touch or running). The smaller fraction of integrative cells in L5 is in line with the suppression of L5 neuron activity with continuous touch. L2/3 neurons were also better at encoding information about run speed in the presence of the textured wall. This is explained by the abundance of integrative cells in L2/3, which display improved encoding of run-speed during wall touch.

Overall, these findings provide direct evidence for a more integrative role of superficial neurons during sensory processing in S1, and suggest that L2/3 neurons play the primary role during contextual modulation of cortical activity and multimodal integration of sensory inputs. Our results indicate that L2/3 neurons enter a continuous monitoring mode during running while L5 neurons mainly respond to salient changes. We therefore speculate that L2/3 and L5 neurons might contribute differentially to behavioral tasks with different requirements. For example, L5 neurons might be more involved in obstacle detection, requiring immediate changes in the motor program, while activity of L2/3 neurons might be critical for texture-discrimination during navigation. In future experiments, it will be interesting to verify such distinct task involvement of L2/3 and L5 populations.

## Methods

**Virus injections and cranial window preparation.** All experiments were conducted in accordance with the ethical principles and guidelines for animal experiments of the Veterinary Office of Switzerland and were approved by the Cantonal Veterinary Office in Zurich.

We used 5–9-week-old male C57BL6 mice. Virus injection and cranial window preparation followed a former description[45] and were performed under isoflurane anesthesia (1.5–2%) with body temperature maintained at ~37 °C using a regulated heating blanket and a thermal probe. The eyes of the mouse were covered by Vitamin A cream (Bausch & Lomb) during the surgery. After hair removal and disinfection with ethanol (Alkopads B.Braun), the skin was opened with a scalpel and the exposed cranial bone was cleaned from connective tissue and dried with cotton pads (Sugi). To express the red calcium indicator R-CaMP1.07[43] in cortical neurons, we injected AAV2.1-EF1α-R-CaMP1.07 into barrel cortex (at 3.3 mm lateral and 1.1 mm posterior to bregma). Two injections of 210 nl of viruses

(~1.21 × 10^13 vg/ml) were performed 300–700 μm below the pial surface to achieve expression in both deep and superficial cortical neurons. Afterwards a circular piece of cranial bone (Ø 4 or 5 mm) was removed using a dental drill, leaving the injection sites in the center. A coverglass (Ø 4 or 5 mm, 0.17 mm thickness) was inserted and secured in place by UV curable dental acrylic cement (Tetric Evoflow). In order to ensure reproducible positioning of the mouse by head-fixation under the microscope objective, a small aluminum hook was glued to the skull on the contralateral side of the head with dental cement. The hook was mounted at slightly tilted angle so that the cranial window was nearly perpendicular to the optical axis of the microscope objective. Intrinsic optic signal imaging was used to verify viral expression area within barrel cortex[45]. The barrel field of single whiskers (mainly B1, B2, C1, and C2) for each mouse was identified under light anesthesia (~0.5–1% isoflurane).

**Tactile virtual reality setup.** Rodents are highly tactile animals. In their natural environment they run through dark tunnels utilizing their whiskers to touch the walls. To simulate this natural behavior, we built a tactile virtual reality setup[65]. Mice were head-restrained on a rung-ladder treadmill (Ø 23 cm) with regularly spaced rungs (1 cm spacing). Run speed and distance were recorded at 40 Hz with a rotary encoder (incremental 5 VDC 360, RI32-O/360AR.11KB, Hengstler). Textures (sandpapers of various graininess: P100 or P1200) were presented on rotating cylinders and were brought in reach of the whiskers with a linear motorized stage (Zaber T-LSM050B stage with built-in controller). Texture contact was established after mice had run a predefined distance on the treadmill. This distance (5–50 cm) was determined for each mouse individually, to allow several seconds of recording before texture touch. A stepper motor (Phidgets 3305_5 NEMA-17 Bipolar 20 mm Stepper) was used to control the speed of the texture rotation. The texture speed was either coupled (Closed-loop) or decoupled (Open-loop) to the animal's run speed. All experiments were performed in the dark. Whiskers on the contralateral side of the cranial window were illuminated with 850-nm infrared LED light while being monitored with a CMOS high-speed camera (Optronis, CL600X2) at 200-Hz frame rate. The behavioral setup was controlled by custom software developed in LabVIEW. This software served as the master control unit for controlling and recording behavioral components and triggered whisker monitoring and two-photon calcium imaging.

**Behavioral paradigms and experimental design.** Animals were free to move on the treadmill and did not receive any reward under any condition. After habituation whiskers were trimmed such that they would not contact the treadmill to avoid somatosensory stimulation by the ladder rungs. We considered three behavioral and stimulation conditions (see Results section in main text for details of conditions): (1) 'No-touch' (5 mice, 10 imaging spots; up to three sessions per spot), (2) 'Closed-loop' (5 mice, 12 imaging spots; up to nine sessions per spot), and (3) 'Open-loop' (5 animals, 11 imaging spots; up to four sessions per spot). To control for confounding effects of the sounds due to motor rotation and linear-stage movement, we rotated the textured cylinder and moved the linear stage in 'No-touch' trials as if in 'Closed-loop' condition but kept the textured 'wall' out of reach for the whiskers.

A week after the cranial surgery, mice were first habituated to the experimenter by handling. Once familiar, animals were accustomed to the behavioral setup, where they freely moved on the rung-ladder treadmill while being head-fixed and presented with Closed-loop rotating texture stimuli. After a week of habituation and training we performed two-photon calcium imaging of neuronal populations in superficial and deep layers of S1 barrel cortex (221–664 μm below the pial surface) during several sessions under all three behavioral conditions. Each behavioral session was composed of 20-s long trials with 3–5 s inter-trial intervals. The number of trials in each session varied from 10 to 80. Animals were kept under these recording conditions maximally for 45 min per recording session and about 1.5 h per day in multiple sessions. For each imaging area, we collected data in 1–6 experimental sessions spread over maximally 10 days. For repeated imaging across several days, individual cells were re-identified by their shape and localization relative to the spatial constellation of the cells in the neighborhood within the imaging field.

**In vivo two-photon calcium imaging.** We used a custom-built two-photon microscope of the Sutter Movable Objective Microscope (MOM) type. This system was equipped with galvanometric scan mirrors (model 6210; Cambridge Technology) and a Pockels cell (model 350/80 with controller model 302RM, Conoptics) controlled by HelioScan software[66]. The objective was a water immersion ×16 objective (CFI LWD 16×/0.80; Nikon). For excitation of R-CaMP1.07, we used a ytterbium-doped potassium gadolinium tungstate (Yb:KGW) laser (1040 nm; 2.5-W average power; ~230-fs pulses at 80 MHz; model Ybix; Time-Bandwidth Products). Fluorescence was collected through a red emission filter (610/75 nm; AHF Analysentechnik) and detected by a GaAsP PMT (Hamamatsu H10771P-40 SEL). We performed in vivo calcium imaging using 33–160 mW average power below the objective for focal depths ranging from 221 to 664 μm. Image acquisition rate was 10 Hz over a 140-μm by 180-μm field of view.

**In vitro patch clamp recordings and calcium imaging.** To compare action potential-evoked calcium dynamics in pyramidal neurons of L2/3 and L5 we performed simultaneous calcium imaging and whole-cell patch clamp recordings in acute brain slices prepared from 2–4 month old wild-type mice. In brief, mice were deeply anesthetized with isoflurane and decapitated. The brain was rapidly removed and put into ice-cold cutting solution containing (in mM): sucrose (189), $NaHCO_3$ (26), $NaH_2PO_4$ (1.2), KCl (2.5), $CaCl_2$ (0.1), $MgCl_2$ (5), and dextrose (10), equilibrated by continuous flow of carbogen (95% $O_2$, 5% $CO_2$). The brain was then trimmed and glued on the stage of a Leica VT1000 vibratome. In total, 300-μm thick coronal slices containing S1 were cut and allowed to recover for 30 min at 34 °C in artificial cerebrospinal fluid (ACSF) containing (in mM): NaCl (125), KCl (2.5), $NaH_2PO_4$ (1.25), $MgCl_2$ (1), $CaCl_2$ (2), $NaHCO_3$ (25), dextrose (10), pH 7.3 with continuous flow of carbogen. After recovery, slices were stored at room temperature for at least 1 h before recording and used within 5 h after cutting. For recording and imaging, slices were transferred to the stage of an upright two-photon microscope (Scientifica Slice Scope) equipped with a ×60 objective and continuously superfused with ACSF (30 °C, 3 ml/min).

Whole-cell patch clamp recordings were obtained from visually identified pyramidal neurons in L2/3 and L5, respectively. Thick-walled patch pipettes (Harvard Apparatus) were pulled with a Sutter-P87 pipette puller and filled with an internal solution containing (in mM): K-Gluconate (135), NaCl (8), HEPES (10), MgATP (2), Na-GTP (0.3), pH 7.3, ~300 mosm. Pipette resistance was 4–7.5 MΩ in the bath and recordings were corrected for the liquid junction potential. Access resistance ranged from 30 to 35 MΩ and was continually monitored for consistency. Voltage recordings were acquired in current clamp mode with an Axon 700B amplifier and digitized with an Axon Digidata 1550A digitizer controlled by the Axoclamp Software version 10.6. Signals were sampled at 10 kHz and Bessel-filtered at 1 kHz. Single or trains of action potentials were induced by 2-ms current injection pulses (3.5 nA) delivered at 50 Hz through the pipette.

We performed two sets of experiments using the synthetic calcium indicator Cal-520 (AAT Bioquest) and the genetically encoded indicator R-CaMP1.07, respectively. To induce R-CaMP1.07 expression, four mice were injected bilaterally at S1 coordinates with AAV2.1-EF1α-R-CaMP1.07, 2–4 weeks before the experiment. For labeling of neurons with Cal-520, the dye was added to the internal pipette solution (100 μM). S1 pyramidal neurons in L2/3 and L5 were identified based on their distance to the pial surface. For Cal-520 experiments, cells were loaded through the pipette for at least 10 min before imaging. For excitation of the calcium indicators we used a Ti:sapphire laser (MaiTai) tuned to either 790 nm (Cal-520) or 1020 nm (R-CaMP1.07). Fluorescence signals were collected with GaAsP PMTs through green (525/50) and red (620/60) emission filters for Cal-520 and R-CaMP1.07, respectively. Scanning, image acquisition, and triggering of current injections were controlled by ScanImage software. Images were acquired at a rate of 16.67 Hz with 128 × 128 pixel resolution.

For analysis, somatic regions of interests (ROIs) were manually selected in Fiji software and the extracted fluorescence signals were analyzed using custom MATLAB scripts. After background subtraction (estimated as bottom 5th-percentile value across the entire movie) calcium signals were expressed as relative percentage fluorescence change $\Delta F/F = (F - F_0)/F_0$, where the baseline $F_0$ was calculated as the mean of the initial 300-ms period of the recording. To correct for fluorescence bleaching apparent in the slice experiments, we fit an exponentially decay function to the $\Delta F/F$ trace, excluding an ~1.7-s period ($-0.24$ to $+1.44$ s) surrounding the peak of the calcium transient and subtracted this fit from the calcium trace. Decay time constants ($\tau$) were obtained from exponential fits to each calcium transient starting from the peak of the transient.

**Slice histology and confocal microscopy.** After the last in vivo experiments mice were anesthetized by i.p. injection of ketamine (0.15 mL, 50 mg/mL). In total, 0.05 mL heparin was injected in the left hearth ventricle and the animal were intra-cardially perfused with 20–25 ml of phosphate buffer (0.1 M, pH 7.3, room temperature) and subsequently with 20–30 -ml of paraformaldehyde solution (PFA; 4% in 0.1 M PBS, pH 7.3, room temperature) both at 11 ml/min. The brain was extracted and postfixed in 4% PFA at 4 °C overnight. Afterward, it was rinsed three times with phosphate buffer and preserved in 30% sucrose (0.1 M phosphate buffer) at −20 °C until further processing. After unfreezing the brain was cut into 50-μm free-floating coronal slices with a microtome (Leica VT1000 S). Brain slices were mounted on microscope slides, embedded in Fluoromount (Dako), and covered by a glass cover. We acquired fluorescence stacks (3-μm z-steps) of R-CaMP1.07 expression with a confocal laser-scanning microscope (Olympus FV1000; 546-nm excitation wavelength).

**Calcium imaging data analysis.** Frames in the time series of two-photon imaging data were registered using a Hidden-Markov-Model, line-by-line motion-correction algorithm[67]. Regions of interests (ROIs) corresponding to individual neurons were manually selected from the mean image of a single-trial using Fiji software[68]. Image frames of other trials were then realigned to the reference trial, from which the ROIs were selected, to account for possible shifts of the field of view throughout an imaging session. R-CaMP1.07 fluorescence signals and behavioral data were analyzed using custom MATLAB scripts (Mathworks). Background fluorescence (estimated as bottom 1st percentile of the fluorescence signal across the entire movie) was subtracted from all trials. Insufficiently motion-corrected

trials or time periods within a trial were excluded from the analysis. In rare cases, ROIs with large motion artifacts were also excluded. Background- and motion-corrected R-CaMP1.07 signals were expressed as relative percentage change of the fluorescence $\Delta F/F = (F - F_0)/F_0$, where baseline $F_0$ was calculated as the 1st per-centile of the smoothed fluorescence trace (51-point 1st-order Savitsky-Golay filter) after concatenating fluorescence signals over all trials within a session. Smoothing over a large window aimed to estimate the baseline fluorescence of a neuron when it was inactive, likely not firing action potentials. Similarly we defined the baseline noise, $\sigma$, as the standard deviation of the fluorescence change during the least noisy 5-s period within a session (1st percentile). We used $\sigma$ for identifying neurons responsive to salient events (e.g., whisking onset, running onset, touch or pertur-bation). Assuming linear summation and single action potential-evoked R-CaMP1.07 $\Delta F/F$ changes of ~8–11% amplitude with ~0.3–0.4 s decay time constant (Supplementary Fig. 3; ref.[44]), a 50% steady-state $\Delta F/F$ change reflects a firing rate change of 11–21 Hz[69], whereas sharp large calcium transients indicate the occurrence of bursts with variable numbers of action potentials. For each neuron, the signal-to-noise ratio was defined as the 95th percentile of $\Delta F/F$ signals recorded during a whole session divided by baseline noise in the same session[46].

**Whisking and running analysis.** We monitored whisker motion at 200 Hz and measured the mean whisker angle across all imaged whiskers using automated whisker tracking software[70]. Whisker angle traces were down-sampled to the two-photon imaging frame rate of 10 Hz. We assigned whisking and no-whisking periods based on whether the standard deviation of the mean whisker angle trace (calculated in a 51-point sliding window) was above or below a predefined threshold (>2.5°). An encoder recorded run speed at 40 Hz and the run speed trace was smoothed with a 1-s Gaussian filter and also down-sampled to 10 Hz. To assign running versus resting periods, run speed traces were further smoothed with a 1.1-s broad 1st-order Savitsky-Golay filter and periods with absolute values of run speed >0.8 cm s$^{-1}$ were considered as 'running periods'.

We analyzed modulatory effects of whisking and running on barrel cortex activity under 'No-touch' condition. To determine the effect of whisking alone we excluded any running period. We defined the whisking modulation index (MI) as

$$\text{Whisking MI} = \frac{\left(\Delta F/F_{\text{rest/whisk}} - \Delta F/F_{\text{rest/nowhisk}}\right)}{\left(\Delta F/F_{\text{rest/whisk}} + \Delta F/F_{\text{rest/nowhisk}}\right)} \quad (1)$$

where $\Delta F/F_{\text{rest/whisk}}$ and $\Delta F/F_{\text{rest/no whisk}}$ denote the mean $\Delta F/F$ value during corresponding resting/whisking and resting/no-whisking states, respectively. Similarly, the effect of running was assessed by comparing mean $\Delta F/F$ values during running versus resting states independent of the whisking state of the animal. Runns to whisking MI as

$$\text{Running MI} = \frac{(\Delta F/F_{\text{run}} - \Delta F/F_{\text{rest}})}{(\Delta F/F_{\text{run}} + \Delta F/F_{\text{rest}})} \quad (2)$$

where $\Delta F/F_{\text{run}}$ and $\Delta F/F_{\text{rest}}$ denote the mean $\Delta F/F$ values in the respective time windows. Neuronal speed tuning was computed similarly to previous studies[3,4]. Run speed traces were binned such that each speed bin covered equal amounts of time, with the exception of the stationary bin (<1 cm s$^{-1}$). Neurons were considered run speed modulated if the mean calcium signal across bins were significantly different (One-way ANOVA, $p < 10^{-3}$). To compute run speed tuning curves we excluded the stationary bin and the rest of the data were fitted by the following equation:

$$y(s) = y_{\text{max}} \exp\left(-(s - s_{\text{max}})^2/\sigma_s\right) \quad (3)$$

where $s$ is the run speed ($s > 1$ cm s$^{-1}$), $s_{\text{max}}$ is the run speed that elicits highest response and $\sigma_s$ is the width of the Gaussian curves such that it equals to $\sigma_-$ for $s < s_{\text{max}}$ and $\sigma_+$ for $s > s_{\text{max}}$. $s_{\text{max}}$, $\sigma_-$, and $\sigma_+$ were free parameters to be fitted. All tuning data were fit by three curves by adding constraints on $s_{\text{max}}$ (i) monotonically increasing function required $s_{\text{max}}$ to be larger than animal's maximum speed; (ii) monotonically decreasing function required $s_{\text{max}} \leq 1$ cm s$^{-1}$; (iii) in band-pass function $s_{\text{max}}$ was not constrained. These three curves were fit on 80% of the data, and we tested the fraction of explained variance of the firing rate on the remaining 20%. For the analysis of the distribution of run speed tuning preferences, we only considered where the proportion of explained variance was at least 30%. A neuron's run speed response was considered band-pass only if the variance explained by the band-pass curve was greater than both a monotonically increasing or decreasing curve and when $s_{\text{max}}$ was >1 cm s$^{-1}$ and smaller than the maximum run speed in the corresponding session.

**Event detection and signal alignment.** For detection of the whisking and running onsets we used the binary running/resting and whisking/no-whisking vectors. Any at least 400-ms long period of no-whisking (or resting) period followed by at least 400-ms of whisking (or running) was detected as a whisking onset (or running onset, respectively). To capture the time course of whisking-evoked $\Delta F/F$ responses independent of running we considered onset-aligned $\Delta F/F$ traces from 2 s before whisking onset until either whisking stopped or the animal started running. Similarly, running-evoked $\Delta F/F$ responses were considered from 2 s before running onset until the animal stopped running. As animals were almost always whisking

when they were running, it was not possible to isolate running from whisking. Because of uncertainties in determining the first actual texture-whisker touch moment, we defined touch onsets as the time points where the linear stage carrying the texture started to move toward the whiskers. The intact set of whiskers together with whisker motion prevented us from detecting the precise moment of first touch of the whisker corresponding to the imaged barrel column. Our rough estimate of the actual time of first touch (of any whisker) is at $158 \pm 48$ ms after the texture cylinder started moving in (reaching its final position to $245 \pm 44$ ms; mean $\pm$ s.d.). We also controlled for potential responses to sounds made by the linear stage or the texture rotation by moving and rotating the texture in 'No-touch' conditions similar to 'Closed-loop' condition but keeping the texture out of reach of whiskers. Both L2/3 and L5 population responses, aligned to the texture moving in, were minimal. Only 3–5% of neurons (16/342 and 5/168 for L2/3 and L5, respectively) were classified as responsive with the same criteria used during touch response analysis.

Finally, perturbation events occurred at the beginning of the 2-s window when texture rotation was halted. To compute the perturbation response we consider only events that occurred when the mean run speed of the animal was larger than $2 \, \text{cm s}^{-1}$. This choice of threshold ensured that a mismatch between run-speed and tactile flow was indeed imposed by the perturbation.

We averaged event-aligned $\Delta F/F$ traces across trials (smoothed with 1st-order Savitsky-Golay filter, 500-ms window) and calculated the event modulation as the difference in mean $\Delta F/F$ values between pre- and post-event windows. For whisking, running and touch modulations we calculated the difference in as the max $\Delta F/F$ in the post-event window minus the pre-event mean $\Delta F/F$. Perturbation modulation was computed as the difference of the mean $\Delta F/F$ values in both pre and post-event windows. Neurons with modulations larger than twice the baseline noise ($2\sigma$) were considered responsive. Specific calculations for each event type are as follows:

For Open-loop condition touch events were divided into 'touch during running' and 'touch during resting' groups. For each event we computed the mean running speed within a 5-s window ($-1$ to $+4$ s) around the touch onset. Events with mean speed $>2 \, \text{cm s}^{-1}$ were selected as 'touch during running' events and those with speed $<1 \, \text{cm s}^{-1}$ were considered 'touch during resting'.

If a neuron was imaged over several sessions we randomly selected a session and considered each neuron once in any population analysis. Only for perturbation responses we selected the session the neurons was most responsive. Hence our perturbation results are an overestimation.

**Functional classification of neurons**. For functional classification of neurons we considered only Open-loop sessions as they comprised various combinations of stimulation and locomotion behavior. Calcium signals were smoothed with a 1st-order Savitsky-Golay filter with 500-ms window and resampled 100 times by selecting equal number of trials of the original session by replacement. For each selection of trials mean $\Delta F/F$ and standard deviation were calculated for four different stimulus-behavior conditions: no-running/no-touch; running in the absence of wall touch; wall touch without running; and concurrent running and wall touch. Cells which show the largest activity in any of these categories were labeled as stationary, run, touch and integrative cells, respectively. Cells were initially assigned to the category where their mean $\Delta F/F$ was the largest. But if the difference in mean $\Delta F/F$ between first and second largest response was smaller than one standard deviation of the largest response category a cell was assigned to the second largest group. The aim of this procedure was to assign cells to single component category (run or texture touch) unless their response significantly increased with the addition of the second component. Final classification of cells, for the first time they were imaged, is shown in Fig. 6b.

This analysis involved four animals, 24 sessions, 11 different imaging areas of 574 cells (338 L3 and 236 L5). Two sessions were not included where one of these four categories were not realized (i.e., there were no periods of texture touch without running). Statistics of cell category distribution was performed by randomly selecting 11 of 24 sessions 10 times. For each selection percent cells in each category was calculated separately for L2/3 or L5 populations. Ten selections provide mean and standard deviation of percent cell category distributions. For each category, two-sampled $t$-test was performed between percent representations in L2/3 and L5 populations.

**Mutual information (I) calculation**. We calculated mutual information to estimate the information about texture speed and run speed represented in the calcium responses of single neurons that were imaged during the Open-loop condition. We included 19/26 sessions where mice spent at least 10% of the time running (or resting).

To do so, we first discretized the $\Delta F/F$ calcium signals, run speed, and texture speed variables into states using uniform binning. The number of bins ($k$) were calculated using Freedman–Diaconis' rule, which aims to minimize the integrated mean squared error of the density estimate:

$$k = \frac{\max(x) - \min(x)}{2.5 \, \text{iqr}(x) \, n^{-\frac{1}{3}}} \quad (4)$$

where $x$ is the vector of values for the variable of interest, $\max(x)$, $\min(x)$, and $\text{iqr}(x)$ are the maximum, minimum, and interquartile range of $x$, respectively, and $n$ is the number of data points for each trial.

Mutual information between $\Delta F/F$ calcium signals, $X$, and variables run speed or texture speed, $Y$, was then estimated with the following equation[71]:

$$I(X;Y) = \sum_{x \in X} \sum_{y \in Y} p(x,y) \frac{p(x,y)}{p(x)p(y)} \quad (5)$$

where $p(x, y)$ is the joint probability function of $x$ and $y$, $p(x)$ and $p(y)$ are the marginal probability distribution functions of $x$ and $y$, respectively.

To account for possible time delays between the calcium signals and variables, we calculated $I(X;Y)$ using time-shifted $\Delta F/F$ data, such that calcium signals were staggered between $\pm 1000$ ms relative to run speed or texture speed. We observed that mutual information peaked when $\Delta F/F$ data were delayed by 200 ms relative to both variables; this delay time was used for all further analyses.

**Encoding of neural activity by behavioral and sensory variables**. To measure how well the calcium signal of a neuron can be inferred from the behavioral and stimulus variables simultaneously acquired in an experimental session, we used a machine learning approach. Specifically we inferred calcium signals from run speed, average whisker envelope and the texture-rotation speed using a random forest algorithm. We used TreeBagger class implemented in Matlab, with 32 trees and the minimum leaf size of 10. To predict calcium signals we used time shifted ($\leq \pm 300$ ms) versions of the sensory and behavioral parameters (also see ref. [72]). The algorithm finds the best mapping between the calcium signal ($y$) and time shifted behavioral or sensory variable ($n$):

$$y(t_k) = f\left(n_{i_{i=1}}^{N} \left(t_{k_{k-p}}^{k+p}\right)\right) \quad (6)$$

where $t_k$ is discretized time (corresponding to the imaging rate); $n_i$ represents the behavioral or sensory variable at time $t_k$ and $N$ is the number of predictors (in this case three for run speed, texture speed and whisk envelope), respectively. Here $p$ is the maximum negative and positive shifts of the calcium signal and it is equal to 3 (corresponding to $\pm 300$ ms window around time $t_k$). The dimensionality of the input variables is $N \times (2p + 1)$.

Prediction of calcium signal was performed separately for each neuron imaged during an 'Open-loop' behavioral session. We only considered time periods during which whiskers were in touch with whiskers The algorithm was trained on a subset of data (the training set, 80%) and evaluated on the remaining test data (20%), which were randomly sampled in 2-s long continuous chunks. We repeated this procedure five times. The quality of prediction was evaluated for each set as the fraction of explained variance

$$Q_i = 1 - \frac{\sum_{t \in C_i} (y(t) - y^{\alpha}(t))^2}{\sum_{t \in C_i} (y(t) - \mu)^2} \quad (7)$$

where $y(t)$ is the calcium signal to be predicted at time $t$, $y^{\alpha}(t)$ is the prediction by the model given sensory and behavioral parameters described above, $C_i$ is the $i$th test set and $\mu$ is the mean of the calcium signal to be predicted of the training set. Any $Q_i < 0$ were replaced with 0. Reported explained variance is $\bar{Q}$ is the mean of all $Q_i$ values.

**Reporting summary**. Further information on research design is available in the Nature Research Reporting Summary linked to this article.

## Data availability
The data that support the findings of this study are available from the corresponding authors upon reasonable request.

## Code availability
Custom-written software code for data acquisition (LABView) and data analysis (MATLAB) is available upon reasonable request.

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

## Acknowledgements

We thank Hansjörg Kasper, Martin Wieckhorst, and Stefan Giger for technical assistance and Yaroslav Sych, Ariel Gilad and Christopher Lewis for comments on the paper. This work was supported by grants from the Swiss National Science Foundation (SNSF) (31003A-149858 and Sinergia grant CRSII3_147660; F.H.), the European Research Council (ERC Advanced Grant BRAINCOMPATH, project 670757; F.H.), Sir Henry Dale Fellowship jointly funded by the Wellcome Trust and the Royal Society (Grant 200501/Z/16/Z; A.B.S.), SNSF Marie Heim-Vögtlin grant (PMPDP3_145476; A.A.), and SNSF Ambizione grant (PZ00P3_161544; A.A.).

## Author contributions

Conceptualization, A.A. and F.H.; methodology, A.A.; investigation, A.A., A.S., M.A.W. and M.H.; software and formal analysis, A.A. and M.H.; writing—original draft, A.A. and F.H.; writing—review and editing, A.B.S., A.S., A.A. and F.H.; funding acquisition, A.A.; resources, A.A. and F.H.; supervision, A.B.S., A.A., and F.H.

## Additional information

**Competing interests:** The authors declare no competing interests.

