## [Peer Review File · Nature Communications]

Reviewers' Comments:

Reviewer #1:

None

Reviewer #2:

Remarks to the Author:

Ayaz et al report on modulation of neural activity in L2/3 and L5 S1 barrel cortex neurons during whisking, whisking + locomotion, and touch under both conditions – additionally in an open-loop and closed-loop configuration with respect to locomotion. I have few complaints with the methodological approach – these minor points are addressed below. My major issue is the lack of substantive new insight provided by this study. The idea of locomotion-associated changes of neural activity has been well established in V1 and A1, and anecdotal reports exist in S1 – although the authors are correct this has not been carefully quantified. Their major finding is a statistically significant, although not particularly large difference in how L5 and L2/3 neurons 'encode' locomotion and touch under the various conditions. The main failing of this study is to connect these results to something meaningful for sensory coding or behavior. The encoding and decoding models do not provide insight. The little bit of data on closed-loop versus open-loop begins to get at this, but the data is still scant. Overall, this study has the sense of being either highly preliminary or secondary to some other data the authors collected. This work is from a highly respected lab that frequently publishes highly impactful and informative studies about neural circuits and coding mechanisms. There is very little of that in this report. It's possible that further analysis of the presented dataset might yield something more interesting – one thing that was not done here is to carefully quantify the amount of touches/force of touches under the various conditions and try to explain the encoding of these variables differentially. Another way to strengthen this study would be to selectively label sub-populations of L2/3 or L5 neurons with defined projection targets and relate the potential differential coding of touch or motor variables to their projection targets.

Minor points:

1. The authors claim they are only labeling L2/3 and L5 pyramidal neurons. There is no direct evidence of this in the study (they cite a previous study of theirs that used a similar vector), and I am skeptical. The promoter in their vector is a pan-cellular driver, and is used widely in AAV vectors for expression in interneurons with the DIO construct. They might be correct, but this should be shown through immunohistochemistry or RNA analysis that all labeled cells are glutamatergic. This is important because the diversity they observe in their data could in part be due to labeling some inhibitory neurons that are known to act differently.
2. Why did the authors use R-CaMP? This should at least be justified, when the green reporters have much better SNR. Is it for depth? Plenty of groups have imaged L5 with GCaMP6.
3. The total data set is not on that many neurons for two photon calcium imaging. Still a lot compared to ephys, but it still suggests a preliminary study.
4. The authors report on a transient response in L5 vs. a sustained response in L2/3. Is this real or an artifact of how calcium is handled differentially between these cells, or because the indicator dye acts different in these two cell groups?
5. Fig. 3a, right: what's going on with the L5 Responses? The 'transient' response seems typically preceded by a big increase in calcium signal for most of the bottom neurons that is not clearly locked to any touch or behavioral variable.
6. Previous electrophysiological recordings in L5 have not always shown such transient responses during touch with objects, but typically more sustained responses following a brief high frequency response. Does the dye just not capture the smaller sustained response in these neurons?

Reviewer #3:

Remarks to the Author:

By using recently developed red-sensitive calcium indicator, R-CaMP1.07, Ayaz and colleagues measured locomotion and whisker-related activity of L2/3 and L5 neurons in barrel cortex. They conclude that L2/3 cells are sustained relative to L5 cells' activity that transient. They show that wall touching differently affected L5 cells whose response was transient whereas L2/3 cells had a sustained response.

Comments:

1. The animals were not performing a task. Thus the functions of the different responses is analyzed with a model. The results as shown in supplemental Figs. 7&8 are only marginally supporting different encoding/decoding of neurons in L2/3 vs. L5.
2. Supp. Fig. 7D: the running vs. resting difference between L2/3 and L5 is marked as significant for L2/3 but not significant for L5. Given the small number of observations and their large variability, this conclusion seems weak at best.
3. In Figs. 2,3&4 responses of L2/3 and L5 neurons are superimposed. This gives the impression that the recordings were done simultaneously. It should be stated that this was not the case (or describe the method used to record simultaneously across layers).
4. Fig. 4B&C, please add interval of confidence (or other statistic) along with the response time course of L2/3 and L5 cells.
5. Was the objective lens tilted? If so, please describe. What was the angle of the glass window relative to the animal axis?
6. Were the animals kept in the dark during the session? If not, please describe.
7. What could be the mechanisms that generated selective expression L5&L2/3? Please discuss.
8. Was the imaging obtained from a specific barrel (or between barrels)?
9. Is there a way to record spikes and image calcium from the same cells in L2/3 and L5 to calibrate the system? This may be important if, for example, the system detects single spikes reliably in L2/3 but is less sensitive in L5. Some discussion of this potential issue should be included.

Reviewer #4:

Remarks to the Author:

Summary

The authors studied the integration of locomotion and wall touching in mouse vibrissa cortex with the use of a tactile virtual reality setup and two-photon calcium imaging with R-CaMP1.07 labeling. This study reveals that neuronal activity in L2/3 and L5A neurons is strongly increased by locomotion and that touch responses are sustained in L2/3 neurons and transient in L5A neurons. This represents new and useful information. The authors should address the "Essential" point, potentially with new data and/or analysis, and address the "Clarifications" as well.

Essential issue :

The authors showed the integration of locomotion with wall-touch (Figures 2 to 4) and the run speed tuning of S1 neurons in the absence of wall-touch (Figure S3). However, these two data sets are not integrated. First, it would be important to see DF/F coding running speed and texture speed on a cell-by-cell basis, rather than solely on a population basis (Figures 3 and 4). A two-dimension scatter plot (or even a histogram) with projections on the running and onto the textures axes would be ideal. This would have to be done for E (early) and L (later) periods.

Further to the above point, is the integration of locomotion and wall-touch is dependent on different neuron types, i.e., the MI, MD or BP cell types.

Clarifications

1) P3. The general statement "whisking behavior has been the main focus of studies on sensorimotor integration in vibrissal primary sensory cortex (S1 or 'barrel cortex')^{10–19} should also include the early work "Phase-to-rate transformations encode touch in cortical neurons of a scanning sensorimotor system. J. C. Curtis and D. Kleinfeld, *Nature Neuroscience* (2009)" and review "Neuronal basis for object location in the vibrissa scanning sensorimotor system. D. Kleinfeld and M. Deschênes. *Neuron* (2011)" on this topic.

2) P3) Introduction. The statement "... whisking was found ... to increase thalamic activity should include reference to the works "Vibrissa self-motion and touch are reliably encoded along the same somatosensory pathway from brainstem through thalamus J. D. Moore, N. Mercer Lindsay, M. Deschênes and D. Kleinfeld, *Public Library of Science: Biology* (2015)" and "Parallel thalamic pathways for whisking and touch signals in the rat. C. Yu C, Derdikman, S. Haidarliu S and E. Ahissar E. *PLoS Biology* (2006)".

3) P8, "39% of L2/3 and 45% of L5 neurons were running speed modulated". On page 45 line 955 the number of neurons modulated by run speed is presented as "441/705 for L2/3 neurons and 193/233 L5 neurons", which is 63% and 83%. These two group numbers do not match. Please clarify the claims

4) Figure S3 shows three types of running neurons, i.e., MI, MD and BP, as above. Please further analysis and discussion locomotion and wall-touch integration by these different cell types.

5) Please further analyze and show the relationship between whisk angle and running speed.

6) Can the animal run only forward or also backward on the treadmill? If it can run backward, what is the nature of neuronal encoding of velocity?

7) P16, "Neurons integrating self-motion and touch are more ...", here "self-motion" can be replaced by "locomotion". As "Self-motion" can also refer to the whisker's self sweeping during active sensing, which is not involved in this study.

8) In Figure 1 (c), please use "L5A" to replace "L5" according to the scale bar.

9) P7, the authors write "Whisking barely modulated the mean activity in L2/3 and L5". As they imaged down to 66 μm depth, which is mainly L5A, it will be more appropriate to use "L5A" rather than "L5" here and throughout the manuscript.

We thank the reviewers for their critical comments, which have triggered further experiments and analysis that are now included in the revised manuscript (we have highlighted major changes in red text). We believe that these additions have improved the manuscript and that we have addressed all points raised by the reviewers. Below we list major changes and additions:

- 1) We now provide histological evidence of low expression of R-CaMP1.07 in inhibitory neurons (Supplementary Fig. 1)*
- 2) We have conducted additional experiments in brain slices that confirm similar action potential-evoked R-CaMP1.07 signals in L2/3 and L5 pyramidal neurons (Supplementary Fig. 3)*
- 3) We now present touch onset responses for all cells to further highlight the transient character of L5 responses in comparison to sustained L2/3 responses (Fig. 3d and Fig. 4b,e)*
- 4) We compared onset responses of all neurons to locomotion and touch, highlighting that transiency of L5 responses is limited to touch onset but not to locomotion onset (Supplementary Fig. 7)*
- 5) We added a presentation of the differences of perturbation responses during running vs resting state (Fig 5e,f,g)*
- 6) We now highlight up and down modulation upon perturbation for all cells (Supplementary Fig. 9)*
- 7) We present reverse correlation between locomotion and touch modulations (Supplementary Fig. 8)*
- 8) We compare encoding of stimulus and behavioral variables across populations by analyzing mutual information (Supplementary Fig. 10)*
- 9) We tracked single whiskers using DeepLabCut and compare the contribution of various whisker parameters to calcium signal encoding (Figure R1)*
- 10) We compare touch modulations and sensory-motor integration properties of different classes of run-modulated cells (Figure R2)*
- 11) We present the relationship between run speed and whisking angle (Figure R3)*

Responses to Reviewer #2:

Ayaz et al report on modulation of neural activity in L2/3 and L5 S1 barrel cortex neurons during whisking, whisking + locomotion, and touch under both conditions – additionally in an open-loop and closed-loop configuration with respect to locomotion. I have few complaints with the methodological approach – these minor points are addressed below. My major issue is the lack of substantive new insight provided by this study. The idea of locomotion-associated changes of neural activity has been well established in V1 and A1, and anecdotal reports exist in S1 – although the authors are correct this has not been carefully quantified.

Answer1. *We do consider several of our findings substantive new insights. Not only do we provide a more careful quantification of locomotion-modulation of S1 L2/3 activity, beyond previous anecdotal evidence, in addition our study goes beyond the state-of-the-art (also considering V1 and A1 studies) by exploring how different layers of the sensory areas alter their responses to sensory stimuli in the presence or absence of locomotion. We provide clear evidence for differential processing of sensory inputs in layer 2/3 and layer 5 and we particularly find that superficial neurons are more integrative—co-processing motor and sensory information— compared to deep layer neurons. To our knowledge these are novel findings that have not been described previously for similar behavioral conditions and will be relevant for the broad community interested in cortical function. In the revised manuscript we have edited the main text to better highlight these salient novel aspects and to better convey our ideas of their meaning regarding sensory coding (see below). We also adapted **Figs. 3 and 4** to more clearly show the L2/3-L5 differences.*

Their major finding is a statistically significant, although not particularly large difference in how L5 and L2/3 neurons 'encode' locomotion and touch under the various conditions.

A2. *We are not entirely sure what the reviewer is referring to. One of our major findings is that L2/3 neurons display a sustained response to continuous touch during running whereas L5 neurons respond only transiently after touch onset (Fig. 3). This difference is statistically highly significant. In the revised manuscript, we have now also added several additional data and amended the text to convince the reviewer that this difference is real (see below). A second major finding is that a higher fraction of L2/3 neurons compared to L5 neurons shows integrative features, i.e., highest activation when sensation is combined with motor behavior. Again, this result is statistically highly significant at $p < 0.01$. Nonetheless, we have performed additional analyses to corroborate this finding with further evidence. We have compared mutual information between calcium signals and run speed in the absence and presence of texture touch. L2/3 neurons increased their information content in the presence of texture touch while the presence of texture touch decreased information content in L5 neurons (Supplementary Fig. 10d). This result indicates better coding of a behavioral variable by superficial neurons when two input streams are combined. In addition mutual information analysis led to distinct outcomes for different cell categories which were defined independently in Fig 6.*

The main failing of this study is to connect these results to something meaningful for sensory coding or behavior.

A3. *In our view, the results of our study prompt new conceptual ideas regarding sensory coding. First, they highlight the large modulation of neuronal population activity in a primary somatosensory area in the locomotion state and the additional modulation of activity if then sensory input arrives. Our study therefore emphasizes the necessity to further investigate sensory coding under naturalistic conditions, when the body is actively engaged in sensory sampling. The novel virtual tactile environment we present in our study should be a useful tool for further studies of this kind. Second, our results point to an interesting laminar difference in sensory coding that is contingent on behavioral state. Based on our finding of sustained versus transient touch responses in L2/3 and L5, respectively, and the stronger sensory-motor integrative features of L2/3 neurons, we speculate that the neuronal population in superficial layers 'stay online' during ongoing sensory-motor sampling—essentially continually monitoring the world and presumably matching it with continuous expectations—while neurons in deeper layers, with their connections to subcortical nuclei such as thalamus, striatum, and brain stem, may react to salient, unexpected events in order to convey this information to relevant subcortical regions, in the end to adapt the animal's behavior. While in the old Discussion we had listed multiple possible mechanisms of how differential laminar processing might be implemented, we perhaps failed to clearly describe these more general ideas regarding the meaning of our findings. We therefore now expanded this part in the Discussion (pgs. 20-24) before going into the Discussion of possible mechanisms.*

The encoding and decoding models do not provide insight. The little bit of data on closed-loop versus open-loop begins to get at this, but the data is still scant.

A4. *We believe our open-loop experiments provide valuable data in understanding how S1 encodes motor and sensory variables. Figure 6 conveys the main finding regarding differences in representation of these variables. The majority of L2/3 neurons were most responsive when both sensory stimulation and running occurred jointly. As explained in A2 we now also provide mutual information analysis, which further supports our statement that L2/3 neurons are more integrative. See also our answers A14,15.*

Overall, this study has the sense of being either highly preliminary or secondary to some other data the authors collected. This work is from a highly respected lab that frequently publishes highly impactful and informative studies about neural circuits and coding mechanisms. There is very little of that in this report.

A5. *We disagree with the reviewer. We present a full data set using a novel approach and we present highly significant and relevant data regarding locomotion-induced modulation of sensory processing in barrel cortex. See also our comments above about novelty.*

It's possible that further analysis of the presented dataset might yield something more interesting – one thing that was not done here is to carefully quantify the amount of touches/force of touches under the various conditions and try to explain the encoding of these variables differentially.

***A6.** A more detailed analysis of the whisker touches to the texture surface certainly would be interesting. However, from our videos (e.g. **Supplementary Video 2**) it is hardly possible to extract touch forces on individual whiskers, especially because we kept the full set of whiskers intact in order to have a naturalistic setting and not to induce plasticity effects. That being said we made further efforts to track single whisker from our movies using the newly available DeepLabCut method (Nat. Neurosci. 21:1281, 2018). Unfortunately, for free whisking episodes (in the absence of texture touch) it was not possible to track single whiskers as they came in and out of focus. Hence we restricted our analysis to texture touch episodes of 'Open-loop' sessions. Using this data set along with run speed and texture speed we compared contribution of various whisking parameters (including frequency of stick-slip events, Chen et al., Nat. Neurosci, 18(8):1101, 2015) in explaining calcium signal of each neuron. Unique contributions of whisking related parameters in predicting calcium signals were insignificant (**Figure R1**). This finding was valid for parameters computed from single whisker tracking and all whisker tracking. On the other hand same analysis showed that both run speed and texture speed contributed significantly to predicting calcium signals (**Supplementary Fig. 10a-c**). For the updated manuscript we have not included **Figure R1** as a supplement as it involves limited data set and we may not have been imaging the corresponding whisker barrel in S1.*

Another way to strengthen this study would be to selectively label sub-populations of L2/3 or L5 neurons with defined projection targets and relate the potential differential coding of touch or motor variables to their projection targets.

***A7.** Subdividing anatomically distinct subpopulations certainly is a very interesting suggestion. However, such investigation is well beyond the scope of this study. We believe the current study presents a strong ground work for such future studies, which can explore the specific mechanisms in further detail.*

Figure R1. Single whisker parameters during open-loop sessions perform as good or worse than average whisker angle in explaining calcium signals.

(a) Left panel: Example whisker image frame acquired during an Open-loop session. Right panel: Same image with best whisker tracked (in green) using Deep Lab Cut (DLC) toolbox. (b) Example traces of single whisker (SW) angle (green) acquired using DLC and normalized average whisker angle of all whiskers (AW, in gray) acquired by whisker tracking software (ref. 69) that was used throughout the study. We only considered time periods where texture was in contact with whiskers as continuous tracking of single whiskers was not feasible in the absence of a texture contact as single whiskers came in and out of the field of view. We could track single whiskers only for 10 sessions (4 L2/3 and 6 L5 imaging sessions). (c) We predicted calcium signals of each neuron during texture touch using a random forest algorithm given run speed, texture speed and various whisking parameters (amplitude of single-whisker envelope, frequency of single-whisker stick-slip events, single-whisker angular speed, amplitude of all-whisker envelope) as predictors. Models were trained and evaluated on separate parts of the data set. Then we shuffled each whisking related parameter one by one while keeping other parameters intact and compared quality of predictions to understand contribution of each parameter in predicting calcium signals. Condition I: Full model, II: AW angle envelope shuffled, III: SW angle envelope shuffled, IV: SW speed shuffled and V: SW stick-slip events are shuffled. Comparisons of mean explained variances in five conditions are shown in red for L2/3 and in blue for L5 neurons (errors are \pm s.e.m.). Shuffling of whisker related parameters did not affect fit quality compared to the full model (Student's T-Test, $p(I-II) = 0.9898$, $p(I-III) = 0.9434$, $p(I-IV) = 0.7643$ and $p(I-V) = 0.3309$ for L2/3; and $p(I-II) = 0.9087$, $p(I-III) = 0.8545$, $p(I-IV) = 0.8686$ and $p(I-V) = 0.6648$ for L5).

Minor points:

1. The authors claim they are only labeling L2/3 and L5 pyramidal neurons. There is no direct evidence of this in the study (they cite a previous study of theirs that used a similar vector), and I am skeptical. The promoter in their vector is a pan-cellular driver, and is used widely in AAV vectors for expression in interneurons with the DIO construct. They might be correct, but this should be shown through immunohistochemistry or RNA analysis that all labeled cells are glutamatergic. This is important because the diversity they observe in their data could in part be due to labeling some inhibitory neurons that are known to act differently.

A8. *In our experience expression of calcium indicators in inhibitory subpopulations under EF1a promoter is difficult (we tried this approach for other projects). We now provide further evidence of low expression rates in inhibitory neurons. We injected viral construct AAV2.1-EF α 1-R-CaMP1.07 into barrel cortex of VGAT-CHR2-EYFP transgenic mice, which express EYFP in GABAergic cell population. Histological analysis revealed that only 3.3% of R-CaMP1.07 expressing neurons were GABAergic neurons in L2/3 and only 6.2% in L5 of S1 cortical slices (**Supplementary Fig. 1**). This is consistent with only about a third of interneurons showing expression and confirms that our findings mainly represent pyramidal cell populations in L2/3 and L5.*

2. Why did the authors use R-CaMP? This should at least be justified, when the green reporters have much better SNR. Is it for depth? Plenty of groups have imaged L5 with GCaMP6.

A9. *The main reason to use R-CaMP1.07 was to gain an advantage for imaging deeper in the cortex as red light scatters less in the tissue (Dana et al., eLife 2016). The newest version of red fluorescent calcium indicators have SNRs comparable to GCaMP6 (see our own paper Bethge et al., PloSOne 2017, for R-CaMP1.07 and Dana et al., eLife 2016, for jRGECOs and Tischbirek et al., J. Physiol. 2015, for a synthetic red indicator). These recent papers demonstrate and highlight the benefit of using red indicators for functional imaging in deep layers of neocortex.*

We do not agree with the statement that 'Plenty of groups imaged L5 with GCaMP6'. To the best of our knowledge, prior to 2016 in vivo calcium imaging of L5 neuron somata has been barely reported with any calcium indicator, with the exception of Mittmann et al., 2011 (GCaMP3 with special lasers) and Masamizu et al., 2014 (GCaMP3 in motor cortex). Very few groups have imaged L5 with GCaMP6. Only recently, parallel to our work, Prevedel et al. 2016 and Yang et al., 2016 achieved this at about 500 micron depth using very specialized microscope setups. We would be happy to consider further references the reviewer might want to refer to.

3. The total data set is not on that many neurons for two photon calcium imaging. Still a lot compared to ephys, but it still suggests a preliminary study.

A10. *The number of neurons analyzed (several hundred in each group) is in the same range as for several other prominently published two-photon imaging studies in neocortex (including previous papers from our lab). In addition we imaged these neurons over multiple sessions across several days, but not to over-represent each neuron we chose to present results from single session for each neuron in the main figures. Some of our supplementary figures (**Supplementary Fig. 3i-I, 5, 6**) consider all imaging sessions independently and report results from up to 1800 neuronal measurements. Hence our study is not preliminary at all. Clearly, there are ongoing developments to expand the field-of-views and to increase the number of imaged neurons. However, the respective publications so far mostly are technical demonstrations with weak biological and behavioral aspects.*

4. The authors report on a transient response in L5 vs. a sustained response in L2/3. Is this real or an artifact of how calcium is handled differentially between these cells, or because the indicator dye acts different in these two cell groups?

A11. *There is no evidence for calcium handling is different in L2/3 versus L5 neurons. All information that is available from >20 years of experiments in vitro and in vivo (e.g. Helmchen et al. 1999, Nature Neuroscience; Svoboda et al., 1999 Nature Neuroscience) indicates that action-potential evoked somatic calcium dynamics in neocortical pyramidal neurons is similar in L2/3 and L5 pyramidal neurons in terms of amplitudes and decay*

times (relating to calcium influx and buffering properties). Most recently Masamizu et al., 2014 directly compared calcium transients in L2/3 and L5 neurons in M1 of mice expressing GCaMP3 and reported similar calcium dynamics. Since we agree with the reviewer that this is an important issue, we now provide further evidence from additional experiments. We performed simultaneous patch-clamp recordings and two-photon calcium imaging in acute cortical slices of wild type mice. We compared calcium dynamics using either the synthetic calcium indicator dye Cal-520 or R-CaMP1.07 and confirmed similar (and fast) action potential-evoked calcium transients in L2/3 and L5 neurons (**Supplementary Fig. 3a-h**). Moreover, we also compared baseline noise and signal-to-noise ratio (SNR) of each neuron in our *in vivo* recordings during ‘no touch’ and ‘closed-loop’ sessions (**Supplementary Fig. 3i-l**; same analysis as in Masamizu et al., 2014). Both L2/3 and L5 neurons showed similar distributions. We provide the relevant descriptions and discussion of this new data set in the revised manuscript (Results lines 116-120 and 209-215, Methods lines 616-662).

Another major argument why differences in sustained vs. transient responses are not due to differences in intrinsic cell properties comes from the locomotion-onset responses. Here, responses after locomotion-onset displayed similar dynamics with sustained increases over seconds for both L2/3 and L5 neurons. This is shown in **Fig. 2i** of the manuscript and in addition in **Supplementary Fig. 7**. For side-by-side comparison we also plotted touch-onset aligned responses of all cells in the **Supplementary Fig. 7b**.

5. Fig. 3a, right: what’s going on with the L5 Responses? The ‘transient’ response seems typically preceded by a big increase in calcium signal for most of the bottom neurons that is not clearly locked to any touch or behavioral variable.

A12. The increased calcium signal preceding the ‘transient’ response is due to higher locomotion-related activity in half of L5 neurons. Thus, it is related to a behavioral variable, which is running.

6. Previous electrophysiological recordings in L5 have not always shown such transient responses during touch with objects, but typically more sustained responses following a brief high frequency response. Does the dye just not capture the smaller sustained response in these neurons?

A13. It would be helpful to know, which previous studies the reviewer is referring to. Again, we do not think that there is any major difference regarding dye sensitivity. Our finding is not conflicting with what the reviewer is describing here either. On average L5 neurons show about 20% and 10% increase in $\Delta F/F$ (compared to pre-touch activity) during early and late phase after touch respectively, which indicates low sustained response following a brief high frequency response. The relative nature of the presented $\Delta F/F$ values (compared to either pre-locomotion-start or pre-touch) should be acknowledged, meaning that L5 may very well retain a low continuous firing rate during running. To better present this now we also plotted actual values of calcium signals (pre-activity not subtracted) in **Supplementary Fig. 7a**. This figure clearly shows how cells increase their activity with locomotion and how later texture touch augments this activity, in a sustained manner in L2/3 and transiently in L5.

Reviewer #3 (Remarks to the Author):

By using recently developed red-sensitive calcium indicator, R-CaMP1.07, Ayaz and colleagues measured locomotion and whisker-related activity of L2/3 and L5 neurons in barrel cortex. They conclude that L2/3 cells are sustained relative to L5 cells' activity that transient. They show that wall touching differently affected L5 cells whose response was transient whereas L2/3 cells had a sustained response.

Comments:

The animals were not performing a task. Thus the functions of the different responses are analyzed with a model. The results as shown in supplemental Figs. 7&8 are only marginally supporting different encoding/decoding of neurons in L2/3 vs. L5.

A14. *We would like to highlight again (as in our answer **A4**) that the main finding regarding differences in representation of sensory and motor variables is shown in **Figure 6**. To strengthen our encoding/decoding analysis we used mutual information between neuronal responses and run speed as an alternative analysis (**Supplementary Fig. 10**). Comparing L2/3 and L5, we found that superficial neurons' information content was higher during touch, in line with a better ability to integrate sensory-motor information (see also our answer **A2** and **A15**).*

2. Supp. Fig. 7D: the running vs. resting difference between L2/3 and L5 is marked as significant for L2/3 but not significant for L5. Given the small number of observations and their large variability, this conclusion seems weak at best.

A15. *We have replaced this figure with **Supplementary Fig. 10**, where we characterized how well time-varying behavioral and sensory variables explain neural responses and also computed how well neurons coded for information under different sensory conditions (please also see **A14**). Mutual information analysis (**Supplementary Fig. 10d**) revealed two interesting observations: (1) 'integrative cells' increased their mutual information about run-speed in the presence of wall-touch whereas for 'run cells' this variable decreased both in L2/3 and L5. (2) The overall increase in information content upon touch in L2/3 can be explained by the larger fraction of integrative cells in superficial layers compared to L5.*

3. In Figs. 2,3&4 responses of L2/3 and L5 neurons are superimposed. This gives the impression that the recordings were done simultaneously. It should be stated that this was not the case (or describe the method used to record simultaneously across layers).

A16. *We apologize for the confusion. Imaging in L2/3 and L5 was done in separate sessions (now clarified in lines 119-120).*

4. Fig. 4B&C, please add interval of confidence (or other statistic) along with the response time course of L2/3 and L5 cells.

A17. *In new **Fig. 4d&c** (old **Fig 4b,c**) we show \pm s.e.m. as shading around the population average responses as we have done in **Fig. 2d,g**, and **Fig. 3e**.*

5. Was the objective lens tilted? If so, please describe. What was the angle of the glass window relative to the animal axis?

A18. *The objective was not tilted but the head holder was implanted in a way to slightly tilt the animals' head when head-fixed. We now specify this in line 557-558 of the text.*

6. Were the animals kept in the dark during the session? If not, please describe.

A19. *Yes, all experiments were performed in the darkness, but in the presence of infrared light source (850 nm) to image the whiskers. This is stated in lines 574-575 of the text.*

7. What could be the mechanisms that generated selective expression L5&L2/3? Please discuss.

*A20. It is a well-known phenomenon that AAV constructs typically do not infect L4 neurons (for which we provide evidence in **Supplementary Fig. 2**). To our best knowledge, the mechanism is still unknown.*

8. Was the imaging obtained from a specific barrel (or between barrels)?

A21. We did not restrict our imaging windows to any specific barrel column. However, we did select our field of views in the region of the barrels that were touching the texture stimulus, which were identified by intrinsic signal imaging (Methods, lines 562-565).

Is there a way to record spikes and image calcium from the same cells in L2/3 and L5 to calibrate the system? This may be important if, for example, the system detects single spikes reliably in L2/3 but is less sensitive in L5. Some discussion of this potential issue should be included.

*A22. For L2/3 neurons expressing R-CaMP1.07 we have previously performed juxtacellular recordings in vivo simultaneously with two-photon imaging and thereby characterized the sensitivity of R-CaMP1.07 (Bethge et al., 2017 (ref. 44); see also the relevant Methods section, page 32). In addition, we have now performed simultaneous patch-clamp recording and two-photon imaging in acute S1 (**Supplementary Fig. 3**). As explained in our response in **A12** the action potential-evoked calcium transients in the somata of L5 neurons are similar to L2/3 neurons. In addition **Supplementary Fig. 3i-l** show that in our in vivo experiments baseline noise levels as well as SNR are similar for L2/3 and L5 neurons reflecting similar sensitivity for detecting action potentials.*

Reviewer #4 (Remarks to the Author):

Summary

The authors studied the integration of locomotion and wall touching in mouse vibrissa cortex with the use of a tactile virtual reality setup and two-photon calcium imaging with R-CaMP1.07 labeling. This study reveals that neuronal activity in L2/3 and L5A neurons is strongly increased by locomotion and that touch responses are sustained in L2/3 neurons and transient in L5A neurons. This represents new and useful information. The authors should address the "Essential" point, potentially with new data and/or analysis, and address the "Clarifications" as well.

We sincerely thank the reviewer for his/her evaluation and clear suggestions and comments on how to improve our manuscript.

Essential issue :

The authors showed the integration of locomotion with wall-touch (Figures 2 to 4) and the run speed tuning of S1 neurons in the absence of wall-touch (Figure S3). However, these two data sets are not integrated. First, it would be important to see DF/F coding running speed and texture speed on a cell-by-cell basis, rather than solely on a population basis (Figures 3 and 4). A two-dimension scatter plot (or even a histogram) with projections on the running and onto the textures axes would be ideal. This would have to be done for E (early) and L (later) periods.

*A23. We have now included a supplementary figure presenting the suggested comparative plots (see **Supplementary Fig. 8**). These plots revealed inverse correlation of touch onset responses with locomotion onset responses for changes in both early and late windows. This effect is more pronounced for L5 population. We point to these findings in lines 223-225 of the text*

Further to the above point, is the integration of locomotion and wall-touch is dependent on different neuron types, i.e., the MI, MD or BP cell types.

A24. To answer this question we have further analyzed touch responses for the three different classes of run-modulated cells. Considering only cells that were identified to be run-modulated (**Supplementary Fig. 5**) we compared their touch onset responses during 'Closed-loop' condition. All three classes qualitatively showed similar touch responses, with L5 neurons showing transient touch responses. (**Figure R2a-c**).

In addition, we compared the integrative features of these classes. MI and BP run-modulated L5 cells mostly fell into the 'run cell' and 'integrative cell' classes with almost no 'texture cells' (**Figure R2d-f**). Although only a small fraction of L5 cells are MD cells, half of them were 'texture cells'. For L2/3 cells the overall distribution across 'run cells', 'integrative cells', and 'texture cells' was similar but with smaller fractions of 'run cells', especially among the MD subpopulation.

If requested by the reviewer we could include this figure in the supplementary material.

Figure R2. Response properties of run speed-tuned neurons. (a-c) Touch onset responses were computed similar to Figure 3 and 4 for the three groups of neurons displaying monotonically increasing (MI) (a), monotonically decreasing (MD) (b) or band-pass (BP) (c) tuning with increasing speed in the absence of texture stimulus. Touch onset is indicated with a red dotted line. L2/3 neurons are plotted in red and L5 neurons in blue. Shading reflects \pm s.e.m. Only neurons significantly run speed-modulated in 'No-touch' sessions ($p < 0.01$) were considered and their mean touch onset responses during 'Closed-loop' sessions are plotted. Neurons may be represented more than once if they were imaged in multiple sessions. (d-f) Classification of run speed-tuned neurons for MI (d), MD (e) and BP (f) tuning, according to their responses in Open-loop sessions. Cells may be repeated for all panels.

Clarifications

1) P3. The general statement "whisking behavior has been the main focus of studies on sensorimotor integration in vibrissal primary sensory cortex (S1 or 'barrel cortex')10–19 should also include the early work "Phase-to-rate transformations encode touch in cortical neurons of a scanning sensorimotor system. J. C. Curtis and D. Kleinfeld, Nature Neuroscience (2009)" and review "Neuronal basis for object location in the vibrissa scanning sensorimotor system. D. Kleinfeld and M. Deschênes. Neuron (2011)" on this topic.

A25. Thank you for the suggestions. We added these references.

2) P3) Introduction. The statement "... whisking was found ... to increase thalamic activity should include reference to the works "Vibrissa self-motion and touch are reliably encoded along the same somatosensory pathway from brainstem through thalamus J. D. Moore, N. Mercer Lindsay, M. Deschênes and D. Kleinfeld, Public Library of Science: Biology (2015)" and "Parallel thalamic pathways for whisking and touch signals in the rat. C. Yu C, Derdikman, S. Haidarliu S and E. Ahissar E. PLoS Biology (2006)".

A26. Thank you for the suggestions. We added these references.

3) P8, “39% of L2/3 and 45% of L5 neurons were running speed modulated”. On page 45 line 955 the number of neurons modulated by run speed is presented as “441/705 for L2/3 neurons and 193/233 L5 neurons”, which is 63% and 83%. These two group numbers do not match. Please clarify the claims.

A27. We thank the reviewer for pointing out this mistake. We corrected the **Supplementary Fig. 5** by replacing in the legend (line 1104) “441/705 for L2/3 neurons and 193/233 L5 neurons” with “276/705 for L2/3 neurons and 107/233 L5 neurons”, which were the number of neurons considered in the figure.

4) Figure S3 shows three types of running neurons, i.e., MI, MD and BP, as above. Please further analysis and discussion locomotion and wall-touch integration by these different cell types.

A28. Please see our response at **A24**.

5) Please further analyze and show the relationship between whisk angle and running speed.

A29. We further analyzed the relationship between whisker set point (mean whisker position/angle) as well as whisking envelope (amplitude of whisking) and run speed. The results are shown below in **Figure R3**. Whisker set point positively correlated with run speed whereas whisking envelope had a sharp increase for low run speeds but saturated or decreased with increasing run speed. These results, which are based on mean whisking angle obtained from all whiskers, are consistent with single-whisker analysis reported by Sofroniew et. al. in Ref. 26.

Figure R3. Relationship between whisking behavior and run speed. (a) Whisking set point versus run speed of a mouse during a single ‘No-touch’ session (shading indicates \pm s.d.). (b) Same as in a but for all 18 ‘No touch’ sessions. (c) The relationship between whisking envelope and run speed for the same single session as shown in a. (d) Same as in c but for all ‘No-touch sessions’.

6) Can the animal run only forward or also backward on the treadmill? If it can run backward, what is the nature of neuronal encoding of velocity?

A30. Backward movements are usually part of jittery forward-backward movements (as if balancing on the treadmill). Unfortunately backward continuous movements are too rare to perform such analysis.

7) P16, “Neurons integrating self-motion and touch are more ...”, here “self-motion” can be replaced by “locomotion”. As “Self-motion” can also refer to the whisker’s self sweeping during active sensing, which is not involved in this study.

A31. We now replaced ‘self-motion’ with ‘locomotion’.

8) In Figure 1 (c), please use “L5A” to replace “L5” according to the scale bar.

Please see below

9) P7, the authors write “Whisking barely modulated the mean activity in L2/3 and L5”. As they imaged down to 664µm depth, which is mainly L5A, it will be more appropriate to use “L5A” rather than “L5” here and throughout the manuscript.

A32. In our experience and for our preparation 700-µm cortical depth already corresponds approximately to the border between L5 and L6 (please compare coronal slices from L5 and L6 R-CaMP1.07 mice, Bethge et al., 2017). Because we do not have any other marker to specifically identify L5A neurons, we prefer to keep the labels as ‘L5’ and not to make that distinction.

Reviewers' Comments:

Reviewer #3:

Remarks to the Author:

The authors have responded thoroughly to the comments of the referees and the paper is now ready for publication.

Reviewer #4:

Remarks to the Author:

I am satisfied with the changes and congratulate the authors on the quality of their work